# Developmental, regenerative, and behavioral dynamics in acoel reproduction

**Vikram Chandra***†, **Samantha Elizabeth Tseng**†‡, **Allison P Kann**,
**Diana Marcela Bolanos**, **Mansi Srivastava***

Department of Organismic and Evolutionary Biology, Museum of Comparative Zoology, Harvard University, Cambridge, United States

**\*For correspondence:**
vchandra1@fas.harvard.edu (VC);
mansi@oeb.harvard.edu (MS)

†These authors contributed equally to this work

**Present address:** ‡The Rockefeller University, New York, United States

**Competing interest:** The authors declare that no competing interests exist.

## eLife Assessment

Xenacoelomorpha is an enigmatic phylum, displaying various presumably simple or ancestral bilaterian features. This **valuable** study characterises the reproductive life history of Hofstenia miamia, a member of class Acoela in this phylum. The authors describe the morphology and development of the reproductive system, its changes upon degrowth and regeneration, and the animals' egg-laying behaviour. The evidence is **convincing**, with fluorescent microscopy and quantitative measurements as a considerable improvement to historical reports based mostly on histology and qualitative observations.

**Abstract** Acoel worms are an enigmatic and understudied animal lineage. Sparse descriptions suggest a diversity of reproductive anatomies across acoels, and likely a corresponding behavioral diversity. Here, we study the reproductive life history of the acoel *Hofstenia miamia*, an emerging lab-tractable model system. We describe *H. miamia*'s reproductive organs, identifying structures previously unknown in acoels. Following worms from zygotes to adulthood, we find that their reproductive organs emerge in a stereotyped sequence as a function of increasing body size. These organs regenerate in a similar sequence after major injuries and are lost in the opposite sequence during starvation-induced de-growth, suggesting that organ growth may be regulated by a single, size-associated program in all contexts. Studying egg-laying behavior, we find that *H. miamia* lay their eggs through their mouths after loading them into their pharynges. Worms lay eggs for months after a single mating, suggesting long-term sperm storage despite lacking a storage organ. They can also lay viable eggs without mating, indicating a capacity for self-fertilization. Finally, worms assess past and present environmental features during egg-laying decisions, frequently laying eggs in communal clutches. Together, our work establishes foundational knowledge for the study of reproductive development, physiology, and behavior in acoels.

## Introduction

Reproduction is critical for evolutionary success, and animals display great diversity in their reproductive life histories. The anatomical structures and life history strategies employed for reproduction have been relatively well-studied in some animal lineages, such as arthropods and vertebrates (*Davies et al., 2013*). However, a full understanding of reproductive biology is lacking in many animal phyla. For example, Phylum Xenacoelomorpha, which includes acoels, nemertodermatids, and xenoturbellids, is an early branching bilaterian lineage of primarily marine worms whose life histories are largely

**eLife digest** Biology is built on a foundation of natural history: a basic understanding of the anatomy, physiology, and behavior of animals in both laboratory and field settings. Centuries of research have provided extensive knowledge about some organisms, yet many animals –such as acoels – remain largely mysterious.

Acoels, a phylum of mostly marine worms, are likely the outgroup to all other Bilateria. Studying acoels and their relatives is therefore vital for understanding the evolution of animal body plans, organs, and other essential traits. Many acoels also have the remarkable ability to regenerate any missing tissue from almost any starting point, and recent advances have established them as a promising model for studying regeneration.

However, fundamental aspects of their biology – including detailed anatomy, regenerative capacities, and reproductive behavior – still require further investigation. Our knowledge of acoel life histories is limited to a small number of anatomical descriptions of adult specimens collected from the field. Characterizing the anatomy of a model acoel, understanding how its organs grow, scale and regenerate, and documenting its reproductive processes can reveal key elements of its natural history, uncover new biological phenomena and provide a basis for future research.

To address this, Chandra et al. employed modern imaging and behavioural techniques to study the life history of the three-banded panther worm, *Hofstenia miamia*. The researchers first examined the growth and regenerative dynamics of reproductive organs and found that the growth rate of these organs depends on the size of an individual, as well as active mechanisms that coordinate organ scaling. Next, using advanced labelling and imaging techniques, Chandra et al. characterised the reproductive anatomy at cellular resolution and identified structures previously undescribed. Finally, they investigated reproductive behaviour and discovered a new mode of egg-laying: *H. miamia* lays eggs through its mouth. The eggs are deposited in communal clutches, and individuals assess environmental conditions to decide whether – and when – to lay eggs, indicating a more complex neural control of reproductive activity.

Acoels offer valuable insights into early animal evolution and regenerative systems, with *H. miamia* emerging as an important model species. Chandra et al. have established key resources and foundational knowledge for systematic study of this species by elucidating core aspects of reproductive development, regeneration, and egg-laying behaviour. More broadly, these findings provide crucial anatomical and behavioural observations relevant to taxonomy, regenerative biology, evolutionary biology, and neuroscience.

unknown (*Bock, 1923*; *Bourlat and Hejnol, 2009*; *Cannon et al., 2016*; *Kapli and Telford, 2020*; *Philippe et al., 2011*; *Figure 1A*).

Roughly 400 species of acoel worms have been described (*Achatz et al., 2013*; *Jondelius et al., 2011*; *Tyler et al., 2012*). Analyses of histological sections of specimens collected in the field suggest that, with few exceptions (*Crezée, 1975*; *Mamkaev, 1965*; *Raikova et al., 1995*), acoels are simultaneous hermaphrodites. Unlike many marine animals that reproduce sexually through spawning and external fertilization, for example, cnidarians, poriferans, echinoderms, etc. (*Giribet and Edgecombe, 2020*), acoels have been reported to reproduce sexually through internal fertilization (*Achatz et al., 2013*). They exhibit a striking diversity in their reproductive anatomy (*Supplementary file 1*). This anatomical variation likely accompanies variation in reproductive physiology and behavior. While a few species have been cultured in the lab (*Bailly et al., 2014*; *De Mulder et al., 2009*; *Shannon and Achatz, 2007*; *Zauchner et al., 2015*), little is known about how acoels develop their reproductive organs, mate, or lay eggs.

To improve our understanding of acoel life histories, we studied reproduction in a lab-tractable species, the three-banded panther worm *Hofstenia miamia* (*Figure 1B, C*). *H. miamia* is a new research organism with features that allow us to study it in controlled conditions: its life cycle can be closed in the lab (worms develop from embryo to gravid adult in 2–3 months), and there is a growing array of experimental tools to observe and manipulate the worms (*Gehrke et al., 2019*; *Hulett et al., 2023*; *Kimura et al., 2022*; *Kimura et al., 2021*; *Ricci and Srivastava, 2021*; *Srivastava, 2022*). Histology-based work has described the coarse anatomy of reproductive organs in animals in the genus *Hofstenia*

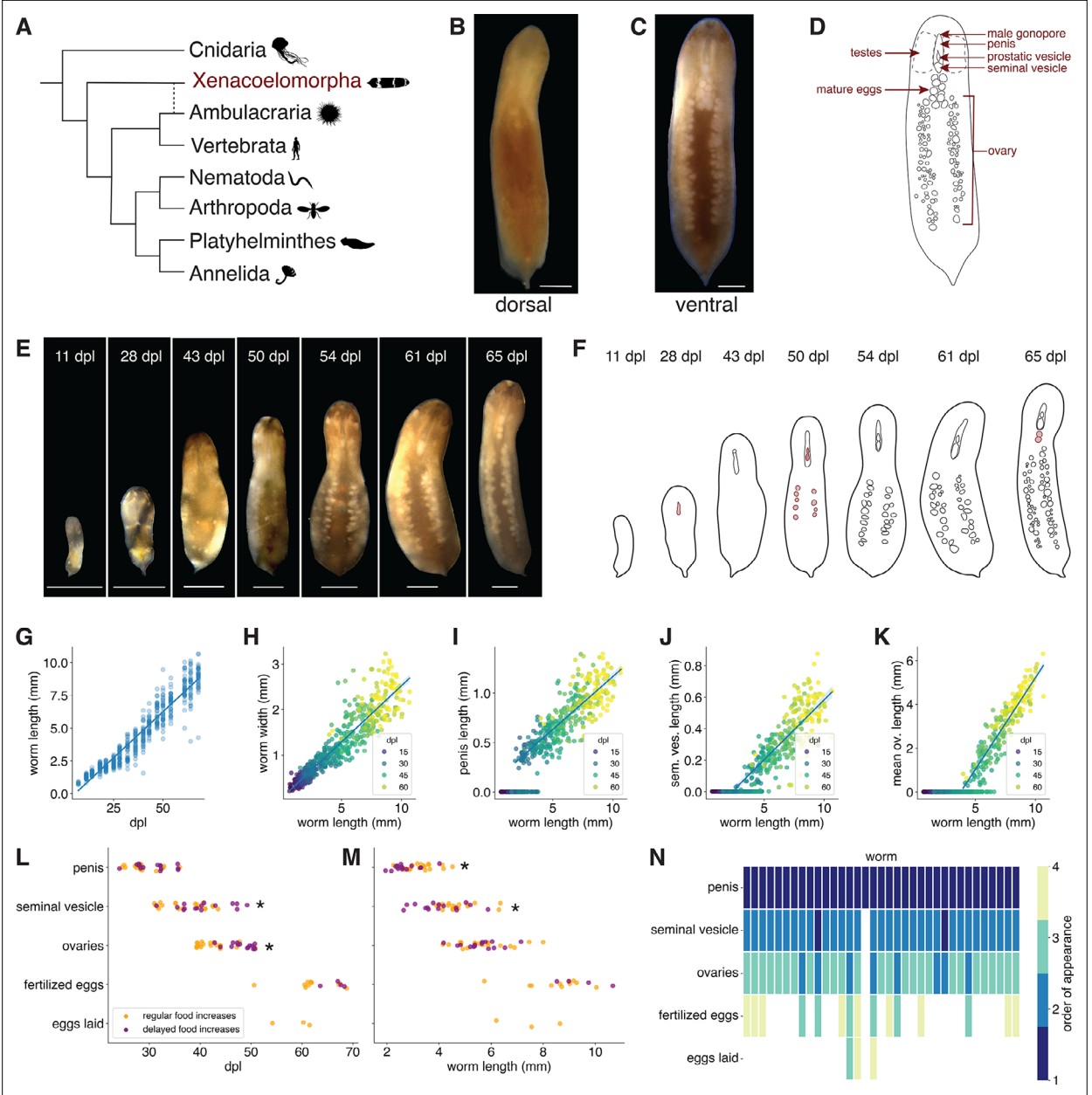

**Figure 1.** Reproductive organs develop in a size-associated sequence. (**A**) Xenacoelomorpha is an early-branching bilaterian lineage of aquatic worms. Animal icons are from phylopic.org, and are in the public domain; the dashed line reflects uncertainty in the consensus phylogeny (*Álvarez-Presas et al., 2024*; *Cannon et al., 2016*; *Kapli and Telford, 2020*; *Philippe et al., 2011*). (**B**) Dorsal view of an adult *Hofstenia miamia*. (**C**) Ventral view of an adult *H. miamia*; most reproductive structures are visible in this view. (**D**) Schematized view of the ventral surface of a worm with known reproductive structures illustrated. (**E**) Time course of a representative worm through development, from hatchling to reproductively mature adult. (**F**) Schematic of time course shown in (**E**) with key reproductive developments illustrated. The first appearance of each organ is highlighted in red. (**G**) The length of worms increases over time ($R^2$ = 0.91, p < 0.0001), and (**H**) worms grow proportionally: their length scales with their width ($R^2$ = 0.85, p < 0.0001). Error band shows 95% confidence interval. (**I–K**) The length of each reproductive organ scales with increases in body size (penis: $R^2$ = 0.70, p < 0.0001; seminal vesicle: $R^2$ = 0.63, p < 0.0001; ovaries: $R^2$ = 0.84, p < 0.0001). Error band shows 95% confidence interval, with zero values excluded from these regressions. (**L**) Worms with delayed feeding increases had significant delays in the appearance of their seminal vesicle and ovaries, but not the penis (Welch's *t*-test for date of appearance for penis: p = 0.08, seminal vesicle: p = 0.04, ovary: p < 0.0001; *n* ≥ 15). (**M**) Worms with delayed feeding increases had a smaller body length when a penis and seminal vesicle appeared, but not when ovaries appeared (Welch's *t*-test for length on date of appearance for penis: p = 0.005, seminal vesicle: p = 0.03, ovary: p = 0.74; *n* ≥ 15). Asterisks indicate statistical significance. (**N**) Ranking the order in which reproductive organs appear (*y*-axis) in developing worms reveals a stepwise pattern of reproductive differentiation. The *x*-axis shows individual worms. dpl = days post laying. Scale bars: 1 mm.

*Figure 1 continued on next page*

*Figure 1 continued*

The online version of this article includes the following figure supplement(s) for figure 1:

**Figure supplement 1.** Gross reproductive morphology of *H. miamia* seen in a ventral view.

**Figure supplement 2.** Growth and scaling patterns in juvenile development.

(*Beltagi and Mandura, 1991*; *Bock, 1923*; *Corrêa, 1960*; *Hooge et al., 2007*; *Hookabe et al., 2024*; *Steinböck, 1966*). However, the fine structure of these organs, how they grow and develop, and how they function in the course of reproductive behavior remains unknown. To address these questions about the reproductive life history of *H. miamia*, we use a combination of confocal microscopy, fluorescence in situ hybridization (FISH), immunofluorescence, and histology, as well as observations of reproductive development and behavior in controlled conditions. We reveal new facets of acoel reproductive biology, show that active processes of growth and destruction ensure coordinated organ development and regeneration, and establish a foundation for the mechanistic study of reproduction in acoels.

## Results

### An overview of *H. miamia*'s reproductive anatomy

The genus *Hofstenia* (*Bock, 1923*) lies within the family Hofsteniidae. The family currently contains four genera (*Ahyong et al., 2024*; *Hookabe et al., 2024*) and is an early-branching lineage within acoels (*Abalde and Jondelius, 2025*; *Jondelius et al., 2011*). Three species are currently recognized within the genus *Hofstenia*: *H. atroviridis*, *H. miamia*, and *H. arabiensis* (*Ahyong et al., 2024*; *Beltagi and Mandura, 1991*; *Bock, 1923*; *Corrêa, 1960*; *Hooge et al., 2007*). Like other acoels, *Hofstenia* are considered to be simultaneous hermaphrodites (*Beltagi and Mandura, 1991*; *Bock, 1923*; *Corrêa, 1960*; *Hooge et al., 2007*; *Steinböck, 1966*). *H. miamia* is likely the most common and widespread species within the genus, with a geographic distribution in the Caribbean and the North Atlantic, including the Bahamas and the Florida coast (*Beltagi and Mandura, 1991*; *Bock, 1923*; *Corrêa, 1960*; *Hooge et al., 2007*; *Steinböck, 1966*). All three *Hofstenia* species are described as having an anterior male reproductive system including a seminal vesicle, prostatic vesicle historically called the *vesicula granulorum* (*Bock, 1923*) or granule vesicle (*Hooge et al., 2007*), a penis with sclerotized needles, 'diffuse' testes, and a ventral male gonopore near the mouth (*Beltagi and Mandura, 1991*; *Bock, 1923*; *Corrêa, 1960*; *Hooge et al., 2007*; *Steinböck, 1966*). The female reproductive system, as described, includes two ovaries spanning the posterior two-thirds of the worm. However, these descriptions were all based on worms roughly 5–8 mm in length (*Beltagi and Mandura, 1991*; *Bock, 1923*; *Corrêa, 1960*; *Hooge et al., 2007*; *Steinböck, 1966*). In our lab cultures, fully grown adult *H. miamia* can be over 1.4 cm in length. We noticed that these large adults were more likely to reproduce than smaller worms, and we reasoned that their anatomy could also be different. We therefore began by re-examining the coarse anatomy of *H. miamia*.

We anesthetized adult worms, mounted them ventral side up on a slide, and imaged them through a dissecting microscope and found, as expected, that *H. miamia*'s male copulatory structures are located in the anterior of the animal, just posterior to the mouth (*Figure 1C, D*; *Figure 1—figure supplement 1A*). These male structures are located within a translucent cylindrical region close to the ventral surface of the worm and immediately ventral to the pharynx. We observed two opaque, white regions within the male copulatory apparatus, located toward its posterior (*Figure 1C*; *Figure 1—figure supplement 1A*). The larger, posterior oval structure corresponds to the seminal vesicle, while the smaller, teardrop-shaped anterior structure corresponds to the prostatic vesicle (*Figure 1C, D*; *Figure 1—figure supplement 1A*; *Beltagi and Mandura, 1991*; *Bock, 1923*; *Corrêa, 1960*; *Hooge et al., 2007*; *Steinböck, 1966*). Previous work reported that *H. miamia* has a penis: a sperm-delivery organ containing rigid needles known as 'stylets' (*Hooge et al., 2007*; *Steinböck, 1966*). Although stylets are difficult to see in this imaging preparation, we identified a small aperture immediately posterior and ventral to the mouth (*Figure 1—figure supplement 1A*), and we observed that the worm extended its penis through it, showing that this aperture is the male gonopore and confirming the location of the penis in the anterior of the worm. As previously described (*Beltagi and Mandura, 1991*; *Bock, 1923*; *Corrêa, 1960*; *Hooge et al., 2007*; *Steinböck, 1966*), we found that the female

reproductive system occupies the posterior two-thirds of the worm (*Figure 1C, D*; *Figure 1—figure supplement 1B*). This system consists of numerous eggs grouped into three distinct clusters: one lateral group on each side of the body extends along the anterior–posterior axis (*Figure 1—figure supplement 1B*), and a medially located cluster lies just posterior to the pharynx (*Figure 1—figure supplement 1C*). The lateral clusters contain oocytes of various sizes, likely corresponding to different stages of maturation. These oocyte-laden regions in the parenchyma have been identified as the ovaries in historical reports (*Beltagi and Mandura, 1991*; *Bock, 1923*; *Corrêa, 1960*; *Hooge et al., 2007*; *Steinböck, 1966*). Posterior to the pharynx, there is often a cluster of uniformly sized, spherical eggs that do not have a germinal vesicle (*Figure 1C, D*; *Figure 1—figure supplement 1C*). The number of eggs at this position is highly variable, ranging from zero to over a dozen eggs. In *H. atroviridis* (*Bock, 1923*), speculated that these eggs are mature, fertilized, and ready to be laid: that is, zygotes. To test whether these eggs were indeed zygotes, we dissected out eggs from this medial cluster, as well as from the lateral ovaries. We found that 46/56 eggs (removed from the medial clusters of 9 worms) developed and hatched into juvenile worms, in line with previously established developmental timing (*Figure 1—figure supplement 1D*; *Kimura et al., 2021*). Meanwhile, all eggs dissected from the lateral ovaries disintegrated within 48 hr. The eggs in this medial cluster are therefore mature, likely fertilized, and ready for laying, while eggs in the ovaries are immature and likely unfertilized. Overall, these observations confirm and extend previous descriptions of *H. miamia* and provide a high-level overview of its reproductive anatomy. Next, taking advantage of *H. miamia*'s tractability, we sought to obtain a finer understanding of its reproductive structures and their development.

## Dynamics of reproductive organ development

How reproductive structures develop as animals reach sexual maturity is not well understood in the Xenacoelomorpha. As in the acoel *Aphanostoma pulchra* (previously *Isodiametra pulchra*) (*Chiodin et al., 2013*), hatchling *H. miamia* have no visible reproductive structures, and previous observations suggested that these develop over a month after hatching as the worms grow toward adulthood (*Kimura et al., 2021*; *Srivastava, 2022*). To determine the timeline and dynamics of reproductive development, we reared 42 zygotes in isolation and monitored their growth over time (*Figure 1E, F*), precisely controlling their rearing environment such that each animal had access to defined amounts of seawater and food (see Methods). To assess whether nutrition affects reproductive development, we split the cohort of hatchling worms into two groups that were fed differently. For each group, we gradually increased the amount of food they were given as they grew larger, to ensure that worms always had ad libitum food access while avoiding unnecessary overfeeding. These food increases occurred periodically, but we delayed them by 2 weeks for one feeding group (*Figure 1—figure supplement 2A, B*). We reasoned that this treatment would delay growth in one group, allowing us to decouple biological age from organ and body growth. Twice weekly, we captured images of the ventral surface of each worm and quantified the size of each visible reproductive structure.

 We found that worms maintained their aspect ratios from hatchling to adult, growing proportionally over time (*Figure 1E–H*; *Figure 1—figure supplement 2C*). At every stage of development, the size of each reproductive organ scaled with body size once that organ appeared (*Figure 1I–K*). Worms with delayed increases in feeding displayed corresponding delays in their growth, both in their body size (*Figure 1—figure supplement 2D*) and in the timing of appearance of their ovary and seminal vesicle (*Figure 1L*). The penis and seminal vesicle developed at a smaller size in these worms compared to worms without delayed feeding increases (*Figure 1M*), suggesting that resource-limited worms may invest disproportionately in male development. Multiple regression found that organ size

**Table 1.** Body length is a better predictor of organ length than age in multiple regression.
Multiple regression coefficients and associated p-values are reported for each organ. Wald's test was used on coefficients to test whether they are significantly different from each other.

| Organ | Day coefficient | Day p-value | Worm length coefficient | Worm length p-value | Wald's test p-value |
|---|---|---|---|---|---|
| Penis | 0.0684 | 0.0002 | 0.1639 | <0.0005 | 0.0082 |
| Seminal vesicle | 0.0425 | 0.0003 | 0.0985 | <0.0005 | 0.0128 |
| Ovary | 0.3361 | <0.0005 | 1.0029 | <0.0005 | <0.0005 |

was explained by both age and body size, but that body size was a better predictor of the size of each reproductive organ (*Table 1*). This suggests that the differentiation and growth of reproductive structures is coupled to the worm's body size, rather than to its age.

We found that reproductive development generally occurs in a stepwise fashion, where organs appear in a consistent sequence at serially increasing body sizes. Newly hatched animals possess all major somatic organs (such as a brain, muscle system, pharynx, and gut; *Kimura et al., 2022*; *Kimura et al., 2021*), but lack all reproductive structures (*Figure 1E, F*). Four to six weeks after egg laying, when the worm is about 3 ± 0.6 mm (mean ± SD) in length, the anterior–ventral tissue housing the future male copulatory apparatus becomes visibly less dense, and a groove becomes apparent in the ventral surface. Soon after, the male gonopore becomes detectable posterior to the mouth. Sperm begins to accumulate in the seminal vesicle around 35 days post-laying, once a worm is about 4.4 ± 0.9 mm (mean ± SD) long. Shortly after, when a worm is 5.6 ± 0.9 mm (mean ± SD), the first oocytes become visible in nascent ovaries. These ovaries grow with the worm over time, with existing oocytes maturing and new ones continuously added. Around 2 months after being laid, typically after a worm is 8.7 ± 1.2 mm in length, eggs appear near the base of the pharynx (*Figure 1E, F, L, M*). These eggs are laid within hours or days, and they develop and hatch into juvenile worms. Given that these worms were isolated since birth, this suggests that *H. miamia* can reproduce through self-fertilization.

To visualize the extent to which reproductive structures develop in sequence, we ranked each structure in each worm by its date of appearance (*Figure 1N*). All worms, without exception, first developed a penis. A few worms simultaneously developed a seminal vesicle; in most other worms, this structure appeared second, with a few developing their seminal vesicle concurrent with nascent ovaries. Ovaries always appeared before the first fertilized eggs were seen in the central region. Next, we asked whether the left and right ovaries of each worm developed in synchrony. We found that on the first day that an ovary was visible, 34/34 worms had both ovaries visible. Partial regression analysis revealed that the sizes of the two ovaries within a worm are strongly correlated, even when correcting for body size (*Figure 1—figure supplement 2E*). Consistent with findings from bilaterally symmetric organs in other animals (*Allard and Tabin, 2009*; *Boulan and Léopold, 2021*; *Harris et al., 2021*; *Vallejo et al., 2015*; *Wolpert, 2010*), this suggests that an active process may synchronize ovarian growth within individual worms.

## De-growth and regeneration of reproductive organs

Next, we asked whether these patterns of growth persisted in different contexts. *H. miamia*, like other studied acoels, is an excellent regenerator and can regenerate all tissues from a wide range of initial tissue configurations (*Srivastava, 2022*; *Srivastava et al., 2014*; *Steinböck, 1966*). We amputated and followed adult worms in three ways: by isolating tail tips (which contain no reproductive structures) and studying how they regenerated the majority of their organs, by cutting worms sagittally and studying how they regenerated their missing half, and by isolating heads from tails and studying how these head fragments regenerated their tails and the posterior region of their ovaries (*Figure 2A*). Head and tail tip fragments both gradually increased in size, with tail tip fragments growing much faster in size (*Figure 2—figure supplement 1A–G*). Within 2 weeks, most of these fragments had regained the characteristic shape of intact worms. Sagittally cut fragments first shrank in size before subsequently growing (*Figure 2—figure supplement 1G*) and appeared to take longer to fully regain normal worm-like appearance (*Figure 2—figure supplement 1C, D*). All reproductive organs gradually regenerated, with growth dynamics contingent on the nature of the injury (*Figure 2B–J*; *Figure 2—figure supplement 1A–F, H, I*). Additionally, we asked how reproductive organs change in another context: degrowth. Anecdotal observations suggested *H. miamia* tolerates long periods of starvation but gradually shrinks in size when deprived of food. We used this starvation-induced de-growth to ask whether reproductive organs scale with decreasing body size. We starved a cohort of adult worms and quantified the de-growth of their reproductive organs over time (*Figure 2K, L*). We found that all worms survived over 3 months of continuous starvation, gradually shrinking over time (*Figure 2M*). Worms maintained their aspect ratios as they shrank (*Figure 2K, L, N*), and their reproductive organs shrank correspondingly (*Figure 2—figure supplement 2A–E*). Partial regression analysis of shrinking ovaries showed that they shrink synchronously (*Figure 2—figure supplement 2F*), suggesting active coordination of de-growth across the left–right axis.

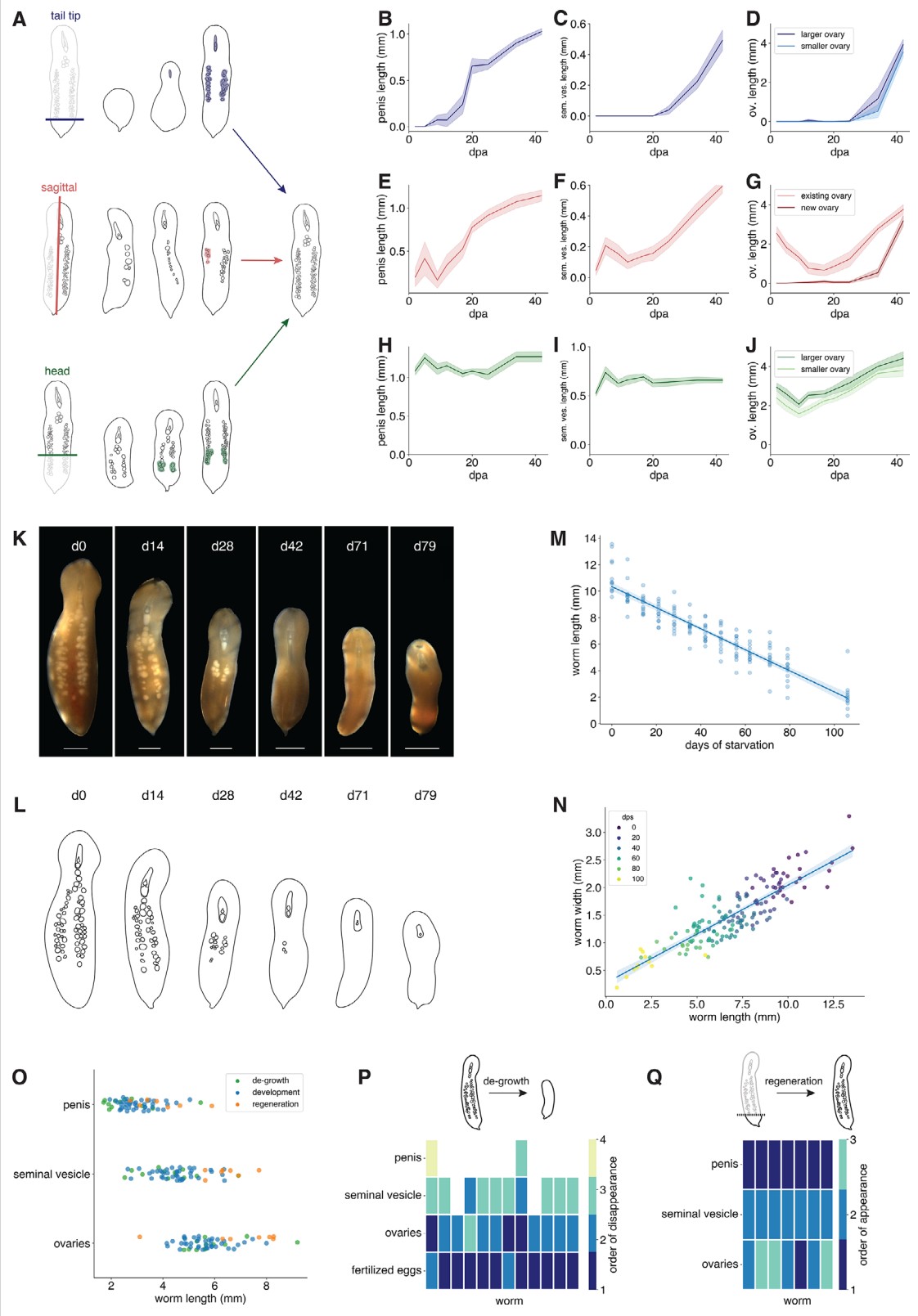

**Figure 2.** Reproductive organ growth follows similar patterns in different contexts. (**A**) Schematic of regeneration of the penis, seminal vesicle, and ovaries following three different amputations. Shading indicates the tissue that regenerates. (**B–J**) Growth dynamics of reproductive organs (within column) for each of three amputations (within row). Error bands show SEM. (**K**) Time course of a starving worm undergoing de-growth and stepwise loss of reproductive organs. (**L**) Schematic of reproductive organ degradation as seen in (**K**) over the course of starvation-induced de-growth. (**M**) Worm

*Figure 2 continued on next page*

*Figure 2 continued*

length decreases over the course of de-growth ($R^2$ = 0.85, p < 0.0001). Error band shows 95% confidence interval. (**N**) Worms shrink as they grow; their lengths and widths decrease proportionally ($R^2$ = 0.73, p < 0.0001). Error band shows 95% confidence interval. (**O**) Across different growth contexts, reproductive organs appear or disappear at roughly consistent body lengths. Ranking the order in which reproductive structures are lost in worms undergoing de-growth (**P**) and gained in regenerating worms (**Q**) shows that organs are gained and lost in roughly the same order in all growth contexts. The *x*-axis shows individual worms in these plots. dps = days post onset of starvation. dpa = days post amputation. Scale bars: 1 mm.

The online version of this article includes the following figure supplement(s) for figure 2:

**Figure supplement 1.** Reproductive system regeneration dynamics from different initial configurations.

**Figure supplement 2.** Across different growth contexts, body and organ scaling rules are conserved.

The stepwise growth relationships we found between reproductive organs and body size during post-embryonic development largely generalized to both regeneration and de-growth (*Figure 2—figure supplement 2G–I*). Starving worms lost most reproductive organs in the opposite order to which they developed them, first losing fertilized eggs, then ovaries, and eventually losing their seminal vesicle (*Figure 2O, P*). However, after 3 months of starvation, most worms continued to retain their penes despite being below the threshold size at which we would expect to see loss of this organ (*Figure 2O*), perhaps because rigid, sclerotized structures degrade differently from soft tissue. In regenerating tail tip fragments, the male reproductive structures generally developed before the female reproductive structures, following similar stepwise patterns to developing juvenile worms (*Figure 2Q*). Scaling coefficients for each organ were statistically distinguishable across development, regeneration, and de-growth, but the effect sizes of the differences between coefficients were generally small (*Figure 2—figure supplement 2G–I*, *Table 2*). Analyzing this effect in more detail, we found that unsurprisingly, amputated worms initially display aberrant organ scaling but recover typical scaling as they regenerate (*Figure 2—figure supplement 2J–L*). Qualitatively, across all of these growth contexts, reproductive organs scaled with body size in similar ways (*Figure 2—figure supplement 2G–I*, *Table 2*).

In sagittally cut worms, each worm fragment retained one of its two ovaries immediately after amputation. These ovaries first partially degenerated before growing back (*Figure 2G*; *Figure 2—figure supplement 1C, D*), consistent with the reported loss of germline tissue during early regeneration in heads regrowing tails (*Hulett et al., 2023*). Body size also decreased after sagittal amputation (*Figure 2—figure supplement 1G*), but ovarian de-growth was disproportionate (*Figure 2—figure supplement 2M*). The rate of ovarian de-growth (normalized to body size) was also significantly greater than the rate of normalized ovarian de-growth during starvation (*Figure 2—figure supplement 2M*). Consistent with studies showing extensive cell death during early regeneration in other systems (*Ballarin et al., 2008*; *Pellettieri et al., 2010*; *Rychel and Swalla, 2008*), this suggests that an active destructive process is responsible for de-growth during ovary regeneration in acoels. The missing ovary began to grow back roughly 1 month after amputation. We found that the ovaries grew asymmetrically in these worms, with the new ovary growing faster (*Figure 2G*). This suggests the existence of an active growth mechanism to ensure that both ovaries reach a symmetric target size.

Together, our results suggest the existence of a size-associated program that regulates the development of reproductive organs, as well as active tissue growth and destruction mechanisms to achieve organ symmetry. It is likely that this program regulates reproductive organ growth in a variety of different developmental contexts, including the transition from juvenile to adult worm, regeneration after injury and tissue loss, and during starvation-induced de-growth. With this high-level understanding of male and female reproductive system anatomy and the scaling relationships that govern their formation in hand, we next sought to understand these systems at higher resolution in order to decipher their functional morphology.

**Table 2.** ANCOVA of the relationship between organ size and body size.

|  | **F-statistic** | **p-value** | $\eta^2$ |
|---|---|---|---|
| Penis | 15.820 | <0.0005 | 0.040 |
| Seminal vesicle | 24.329 | <0.0005 | 0.069 |
| Ovary | 58.905 | <0.0005 | 0.175 |

## Fine structure of the male reproductive system

To understand the organization of male reproductive structures at high resolution, we visualized them using a combination of histology, immunofluorescence, staining with live dyes, and fluorescence in situ hybridization (FISH) (*Figure 3*; *Figure 3—figure supplement 1A*). The most anterior part of the male copulatory apparatus is the penis. This structure is difficult to visualize under reflected white illumination (*Figure 1C*; *Figure 1—figure supplement 1A*; *Figure 3—figure supplement 1A*), but can be seen when worms extend it, which we have observed them do both spontaneously and during mating (data not shown). Staining fixed worms with the dye SiR-actin labels the structure clearly and can be visualized well in smaller adult worms more amenable to confocal microscopy, enabling a high-resolution view of the structure (*Figure 3A–C*).

Components of the penis can also be seen clearly in histological sections of adult worms (*Figure 3D*). These show that the penis contains a bundle of rigid needle-like structures (referred to as stylets) (*Figure 3E–G*, *Figure 3—figure supplement 1B, C*). Our observations of the stylets match the description of sclerotized needles described in other species in the genus and in other studies of *H. miamia* (*Beltagi and Mandura, 1991*; *Bock, 1923*; *Corrêa, 1960*; *Hooge et al., 2007*; *Steinböck, 1966*). The stylets are situated immediately posterior to the base of the penis sheath – a long, conical structure (*Figure 3C*; *Figure 3—figure supplement 1E, F*; *Figure 3—video 1*). The cavity in the anterior of this sheath has previously been referred to as the 'male antrum' (*Beltagi and Mandura, 1991*; *Bock, 1923*; *Corrêa, 1960*; *Hooge et al., 2007*; *Steinböck, 1966*). The base of the sheath is cup-shaped and is lined with posterior-facing hair-like extensions (*Figure 3G*). At its anterior end, it connects to the male gonopore (*Figure 3C*). Both the gonopore and the sheath are lined with cilia (*Figure 3—videos 1 and 2*), consistent with reports in other hofsteniid species (*Beltagi and Mandura, 1991*; *Bock, 1923*; *Corrêa, 1960*; *Hooge et al., 2007*; *Steinböck, 1966*). During mating, the penis stylets are likely pushed anteriorly, through the sheath and out of the male gonopore. This could evert the sheath, resulting in a penis-extension state in which the sheath emerges through the gonopore around the stylet, with the ring of protrusions now near the anterior tip of the penis. It is possible that these protrusions have a sensory function and could inform fine penis movements during mating. Immunofluorescence with an antibody against FMRFamide (see Methods) revealed a set of cells that resemble neurons encircling the base of the sheath (*Figure 3—figure supplement 1E*). Further studies will be needed to understand how these components regulate copulatory behavior.

Next, we focused on the teardrop-shaped prostatic vesicle, which is situated anterior to the seminal vesicle (*Figure 1C, D*) and immediately posterior to the penis stylets (*Figure 3H, I*). Our histological studies show that the prostatic vesicle (*Figure 3—figure supplement 2C–E*) is surrounded by glands (*Figure 3—figure supplement 2F–H*). The secretions of this organ are thought to mix with the sperm as it passes through (*Faubel, 1983*; *Hyman, 1951*). Unexpectedly, our immunofluorescence experiments also revealed a layer of cells enclosing the prostatic vesicle (*Figure 3I*). This layer may be an epithelial tissue most likely associated with the prostatic glands. The interior of the prostatic vesicle contains densely packed sperm cells (*Figure 3D, I*; *Figure 3—figure supplement 2C–G*).

The seminal vesicle is the posterior-most organ of the male copulatory apparatus (*Figure 3J*). Histological staining as well as immunofluorescence staining of the muscle marker Tropomyosin showed that the seminal vesicle and prostatic vesicle are both encircled by layers of muscle (*Figure 3—figure supplement 1F*; *Figure 3—figure supplement 2C–H*), consistent with previous descriptions (*Beltagi and Mandura, 1991*; *Bock, 1923*; *Corrêa, 1960*; *Steinböck, 1966*). Histological and nuclear staining of the seminal vesicle in adult worms (*Figure 3J–L*; *Figure 3—figure supplement 1F*; *Figure 3—figure supplement 2G, H*) reveals that it consists primarily of sperm cells. To confirm this, we dissected the seminal vesicle out of adult worms and stained it with a nuclear dye. We found that individual cells exited the structure over time, and high-magnification imaging showed that they were indeed sperm cells, each with a ~23-µm-long nucleus and a ~22-µm-long flagellum (*Figure 3M*). During one mating event, we observed a failed attempt at insemination by a worm that resulted in the release of a packet of sperm outside its partner. We recovered this 0.6-mm-long packet of sperm (*Figure 3—figure supplement 1G*) and found that it consisted of cells with the same morphology as those in the seminal vesicle (*Figure 3—figure supplement 1H*), confirming that these are indeed sperm. As an aside, acoel sperm has historically been described as 'biflagellate', intended to mean that sperm cells each contain two axonemes, and this property is thought to be a defining synapomorphic character of acoels (*Achatz et al., 2013*; *Petrov et al., 2004*). We show that *H. miamia* sperm have a single

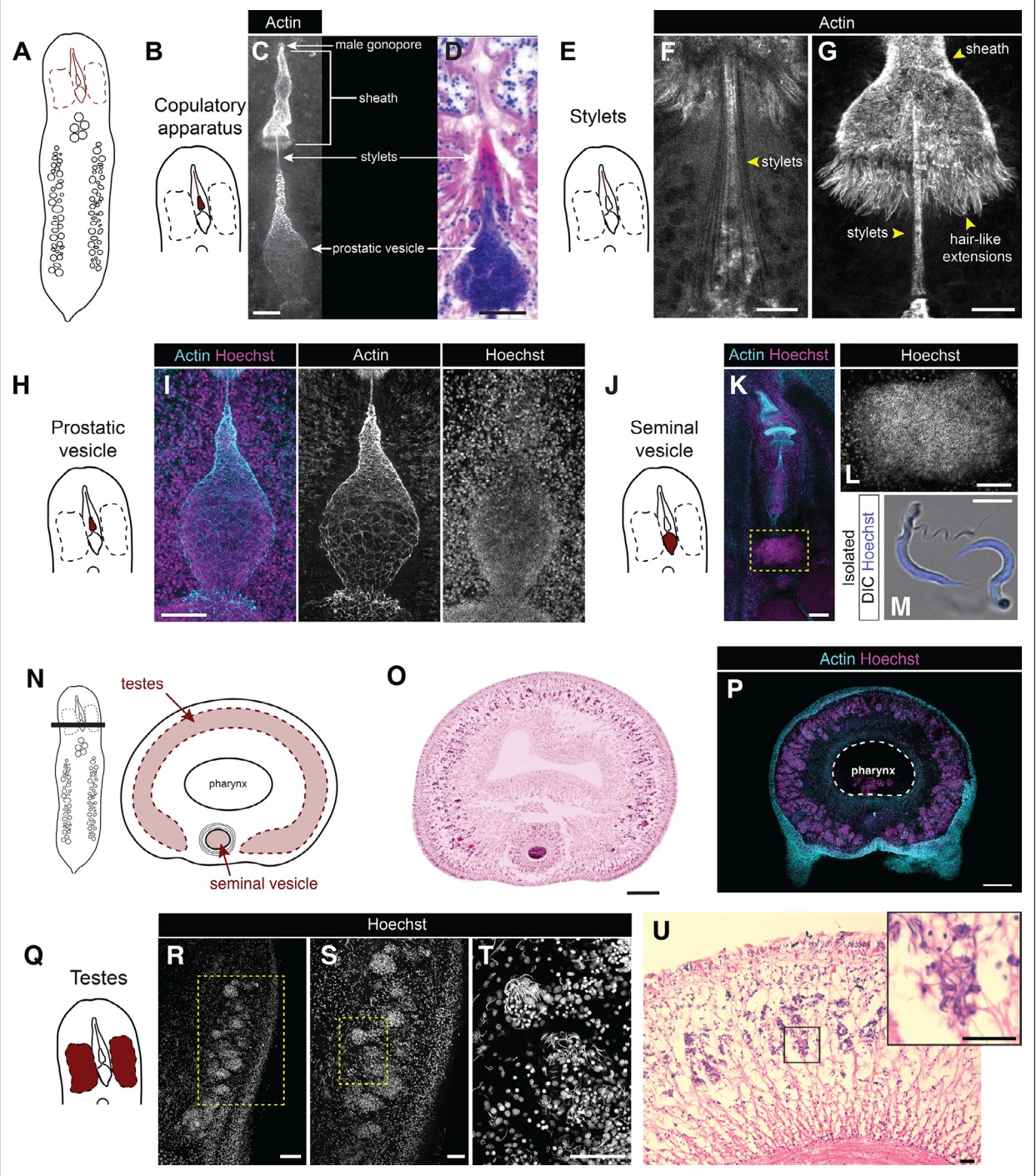

**Figure 3.** Male reproductive anatomy. (**A**) A schematized view of the ventral surface of the worm with male reproductive structures highlighted in red. (**B**) Schematic of male reproductive structures with the copulatory apparatus (excluding the seminal vesicle) highlighted. (**C**) Labeling with an actin dye (white) labels the male gonopore, sheath, penis stylet, and prostatic vesicle. (**D**) A histological section also reveals these organs. (**E**) Schematic of the male reproductive system, with the penis stylet highlighted. (**F**) The stylets are a bundle of needles labeled by actin. (**G**) The posterior of the penis sheath terminates in a ring of hair-like projections, also labeled by actin. (**H**) Schematic of the male copulatory apparatus, with the prostatic vesicle highlighted. (**I**) Actin staining with a nuclear label (Hoechst) shows that the prostatic vesicle is enveloped by a thin epithelium-like layer, and contains densely packed sperm. (**J**) Schematic of the male copulatory apparatus, with the seminal vesicle highlighted. (**K**) The morphology of the copulatory apparatus in mature, adult worms is similar to that of early adults (previous panels). (**L**) The seminal vesicle of this adult worm contains densely packed sperm. (**M**) Dissecting out an adult seminal vesicle allows labeling of individual sperm cells, showing their distinctive morphology. (**N**) Schematic of a transverse view of an adult worm's anterior, showing the relative organization of the seminal vesicle and testes. (**O**) Transverse sections show that testes

*Figure 3 continued on next page*

*Figure 3 continued*

appear as a continuous structure that spans the dorsal surface of the worm. (**P**) The testes extend through the dorso-ventral axis of the worm and wrap around the head. The pharynx (labeled and circled with a dotted line) contains residual food. (**Q**) Schematic of the male copulatory apparatus, with the testes highlighted. (**R–T**) Nuclear staining on an adult worm, cut sagittally, reveals the testes, which contain dense bundles of sperm organized around clusters of cells in the parenchyma. (**U**) Histological sections confirm this organization of the testes. Scale bars: 20 µm (**C, U**), 10 µm (**F–G, M**), 50 µm (**D, I, S, T**), 100 µm (**K, L, R**), 200 µm (**O, P**).

The online version of this article includes the following video and figure supplement(s) for figure 3:

**Figure supplement 1.** Fine structure of *H. miamia*'s male reproductive system.

**Figure supplement 2.** Histology of the male reproductive system.

**Figure 3—video 1.** The penis sheath is ciliated.

https://elifesciences.org/articles/105712/figures#fig3video1

**Figure 3—video 2.** The male gonopore is ciliated.

https://elifesciences.org/articles/105712/figures#fig3video2

flagellum and suggest that the axonemal properties of the flagellum may be more accurate taxonomic characters than external flagellar morphology.

Next, we sought to identify the testes (the organs of sperm production). Consistent with previous observations (*Beltagi and Mandura, 1991*; *Bock, 1923*; *Corrêa, 1960*; *Hooge et al., 2007*; *Steinböck, 1966*), nuclear staining and histological studies of adult worms revealed dense clusters of sperm in the anterior parenchyma, wrapping around the worm's pharynx (*Figure 3N–U*). Tropomyosin labeling showed that the testes are embedded within layers of muscle (*Figure 3—figure supplement 1I, J*). Next, inspecting germline markers identified in previous single-cell RNA sequencing data (*Hulett et al., 2023*), we identified one likely to label testes: the gene *pa1b3-2*. FISH for *pa1b3-2* reveals that this gene does indeed specifically label testes, and that its expression coincides well with the dense cellular clusters visible through nuclear staining (*Figure 4—figure supplement 1A and B*).

Finally, we studied the development of the male reproductive system, using confocal microscopy to visualize the maturation of its components at high resolution. Hatchling worms do not possess any visible reproductive structures. Actin staining reveals that, within the copulatory apparatus, the sheath and stylets emerge first, followed by a gradually inflating prostatic vesicle (*Figure 4A*). The hair-like extensions of the sheath grow and become more conspicuous over time (*Figure 4A*). We used two methods to visualize testes maturation. First, reasoning that nuclear staining may not allow us to visualize testes at the earliest developmental stages, we performed FISH to detect mRNA for *pa1b3-2* in juvenile worms of various sizes. In the smallest worms (i.e., ~<2 mm), there was no expression (*Figure 4B, C*). In larger, reproductively immature worms, we detected expression of *pa1b3-2* in two lateral clusters that expanded slightly along the anterior–posterior axis as development progressed (*Figure 4B, C*). Since adult testes are a single cylindrical structure that wraps around the pharynx, this early developmental pattern suggests that the two lateral structures must grow along the dorso-ventral and medio-lateral axes until they meet in the middle. Second, to test this hypothesis, we performed nuclear staining on transverse sections of worms across a broader section of development (*Figure 4D*; *Figure 4—figure supplement 1C*). In small worms, the testes are lateralized and do not meet on the dorsal surface. In larger worms, these lateral regions have expanded across the dorsal surface of the worm to form a single apparently continuous region (*Figure 4E*). Many acoel species are thought to have paired, lateral testes (*Supplementary file 1*), a feature of taxonomic importance (*Jondelius et al., 2011*). While juvenile *H. miamia* have paired testes, these organs morph into a single cylindrical structure in the transition to adulthood. As lab-reared worms can be substantially larger than those found in the wild, these results raise the possibility that other acoels may undergo similar morphological changes during development.

Together, our data provide a high-resolution view of *H. miamia*'s elaborate male reproductive morphology. Consistent with our quantitative data, we find that this morphology emerges in a stereo-typed developmental sequence during the transition to adulthood in which the penis sheath develops first, followed by nascent testes, the stylets, and the membrane-like structure enveloping the prostatic vesicle. Sperm, presumably from the testes, then travels to the prostatic vesicle. Over time, as all structures grow, the hair-like projections on the back of the sheath become more prominent, the testes extend both along the anterior–posterior axis and dorsally to wrap around the head, and sperm

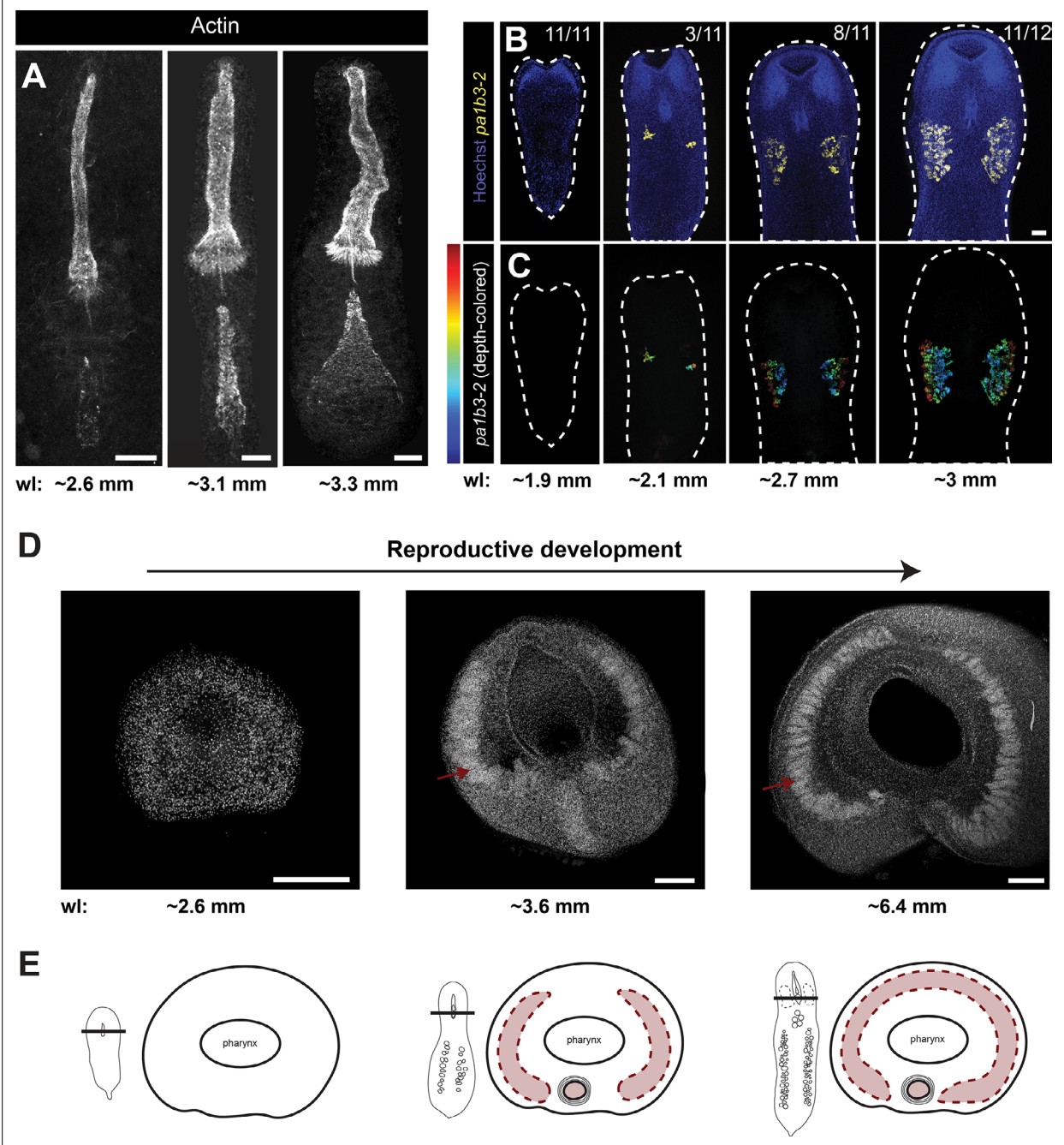

**Figure 4.** Stepwise emergence of components of the male reproductive system. (**A**) Actin-dye labeling shows how the male reproductive system changes over the course of post-embryonic development (shown here from left to right). The sheath and stylet emerge first, followed by the appearance of the prostatic vesicle. (**B**) Fluorescence in situ hybridization (FISH) for the male germline marker *pa1b3-2* results in two regions of ventrolateral expression that extend along the dorsal–ventral axis to different depths. Images are organized by pseudo-time: from least-developed (and smallest) on the left, to most-developed (and largest) on the right. Panels in (**C**) show depth-coloration, showing that the testes extend through the dorso-ventral axis. (**D**) Cross-sections of worms at different points in reproductive development stained with nuclear dye show that testes grow toward the dorsal surface and eventually meet to form one continuous structure. (**E**) Cartoon schematic of cross-sections shown in (**D**). Scale bars: 20 μm (**A**), 100 μm (**B, C**), 200 μm (**D**). Estimated worm lengths (wl) are noted under each panel.

The online version of this article includes the following figure supplement(s) for figure 4:

**Figure supplement 1.** Testes emerge as lateralized regions and grow to span the dorsal surface as worms grow.

cells fill the seminal vesicle. The prostatic and seminal vesicles appear to be surrounded by muscle, epithelia, and gland cells. Our histological sections suggest that at least in adult worms, sperm cells may enter the seminal vesicle and travel forwards into the prostatic vesicle. We do not yet understand how this occurs in juvenile worms, how sperm cells navigate to these vesicles from their varied points of origin, or how they are hypodermically injected during mating.

## Fine structure of the female reproductive system

As observed through imaging of adult worms using a stereo microscope, the female reproductive system in *H. miamia* consists of two lateral ovaries running longitudinally along both sides of the body, and one medial cluster of mature eggs located posterior to the pharynx (*Figure 5A–C*). To visualize the structure of the ovaries, we used FISH to label mRNA of a previously identified germ-line marker: *cgnl1-2* (*Hulett et al., 2023*). We found that this gene specifically labeled the oocytes (*Figure 5D*). Moreover, immunostaining with a custom antibody against Piwi-1 (typically a stem cell and germline marker) also labeled the ovaries (*Figure 5E*). Unlike in many other organisms where *piwi* homologs are expressed in male gonad tissue (*Deng and Lin, 2002*; *Gonzalez et al., 2015*; *Jehn et al., 2018*; *Kuramochi-Miyagawa et al., 2004*; *Miramón-Puértolas et al., 2024*), we did not detect Piwi-1 expression in *H. miamia* testes. Possible explanations include potential low expression of Piwi-1, the use of other Piwi proteins in spermatogenesis, or that male germline progenitors (which could express Piwi-1) may not be co-located with maturing sperm in the testes. We were also able to visualize the ovaries in histological sections (*Figure 5C*). These methods, together with our earlier images of ovaries at different worm ages, showed that oocytes are not organized by maturity within the ovary (*Figure 5—figure supplement 1A*). We did not detect a membrane or lining that envelopes the ovaries. This is consistent with previous work that suggests that acoels lack true ovaries sensu stricto: oocytes appear and develop within the parenchymal tissue without a specialized membrane that forms a discrete organ (*Eckelbarger and Hodgson, 2021*; *Rieger et al., 1991*; *Schmidt-Rhaesa, 2007*).

Oocytes in *Hofstenia* are surrounded by a layer of follicular cells, which are thought to provide nutrition and secrete the eggshell (*Bock, 1923*; *Rieger et al., 1991*). We visualized this cell layer with immunofluorescence and histology and found that it was consistently present around every oocyte in the ovaries irrespective of their developmental stage (*Figure 5F, G*; *Figure 5—figure supplement 1B–D*). We observed that many follicle cells have very large nuclei, perhaps the result of the fusion of neighboring cells (*Figure 5G*; *Figure 5—figure supplement 1B–D*). We also found clusters of sperm in the layer of follicular cells surrounding maturing oocytes of a variety of developmental stages (*Figure 5F, G*, insets), consistent with (*Bock, 1923*) proposal that the follicle may 'trap' sperm and control the fertilization of maturing oocytes. All oocytes contained a granular substance primarily distributed peripherally, and a germinal vesicle occupying a substantial portion of the oocyte (*Figure 5H–J*). A germinal vesicle, a nucleus arrested in prophase I of meiosis, is a feature of oocytes in most animals (*Grossman et al., 2017*; *Herlands and Maul, 1994*; *Liu et al., 2006*; *Munro et al., 2023*). It is broken down during oogenesis or after fertilization (*Falleni and Gremigni, 1990*; *Sagata, 1996*). The presence of a germinal vesicle in the oocytes in *H. miamia*'s ovaries suggests that these oocytes are unfertilized. It is likely that oocytes mature in the ovaries, the germinal vesicle then breaks down, following which oocytes are fertilized. These mature, fertilized eggs may then travel from the ovary to the central cavity prior to being laid.

We could not determine conclusively where or when fertilization happens, or where the egg capsule (chorion) is produced. However, the spatial organization of the female reproductive system suggests that an oocyte must mature, become fertilized, and then be transported to the cavity behind the pharynx. Our internal examinations did not identify any oviducts, canals, or additional structures that could facilitate this migration from the ovaries to the medial cluster. How eggs are transported thus remains unknown.

## Egg-laying behavior

Egg laying has only been directly observed a few times in acoels, mostly in species within the Convolutidae (*Costello and Costello, 1939*; *Gardiner, 1898*). From these observations, acoels are known to lay eggs through the female gonopore if present (*Gardiner, 1898*), a mode of egg laying used by many platyhelminths (*Tong and Ong, 2020*), or through breaks in the body wall (*Apelt, 1969*;

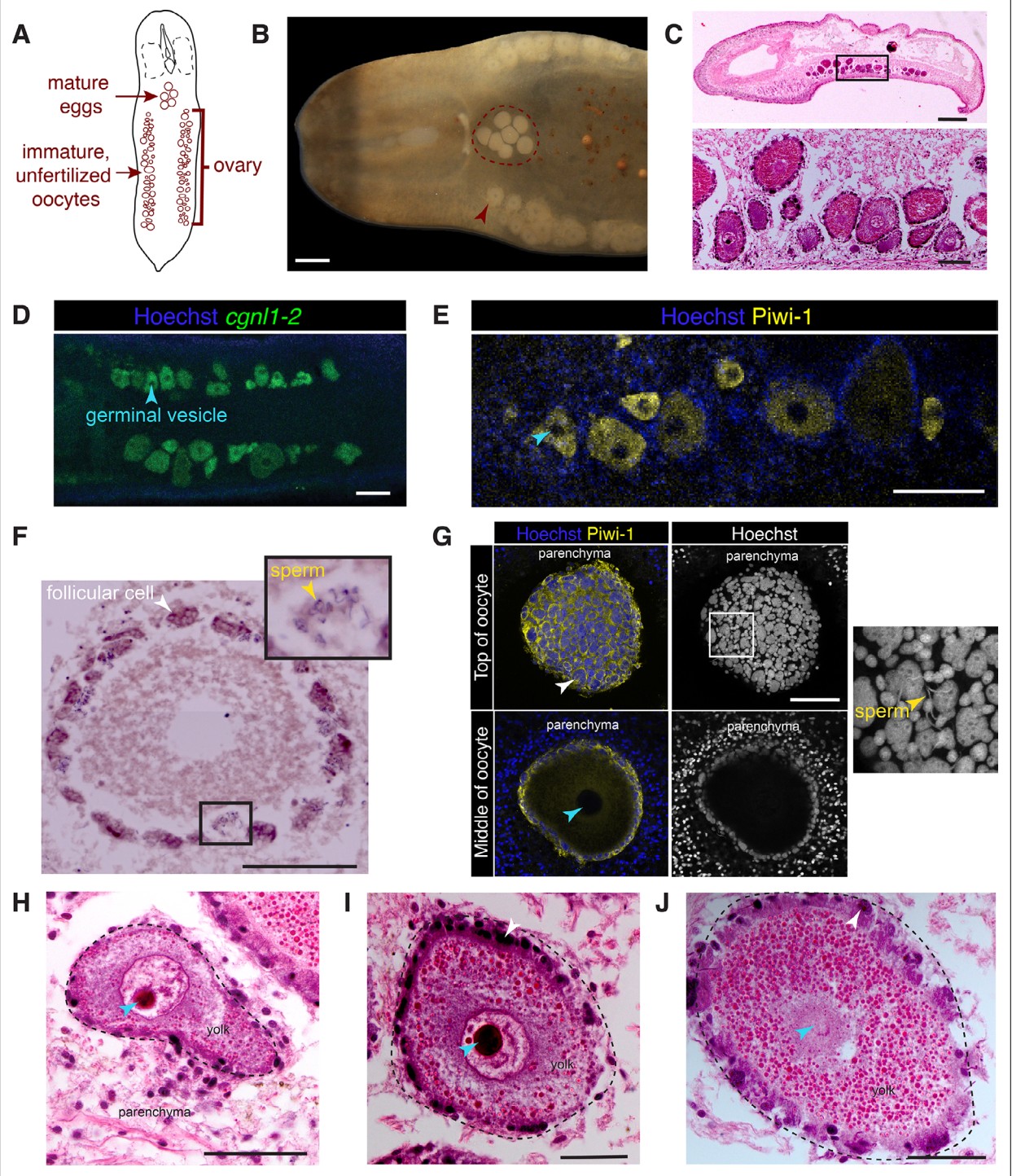

**Figure 5.** Female reproductive anatomy. (**A**) A schematized view of the ventral surface of the worm with female reproductive structures highlighted. (**B**) Eggs near the pharynx of the worm (within the red circle) are fertilized and mature while oocytes in ovaries (red arrow) are immature or unfertilized, with a visible germinal vesicle. (**C**) A sagittal histological section shows that the ovaries contain oocytes of varied size and maturity embedded in the parenchyma. (**D**) Fluorescence in situ hybridization (FISH) shows that *cgnl1-2* labels immature oocytes in the ovaries. (**E**) Oocytes in ovaries are also labeled by a Piwi-1 antibody. (**F**) A histological transverse section of an immature oocyte encircled by follicular cells. Inset: sperm cells appear to be trapped in the follicle. (**G**) Piwi-1 immunofluorescence confirms the organization of follicular cells, and nuclear staining sometimes identifies sperm apparently trapped in its surface (inset). Histology also shows that immature oocytes may have irregular shapes (**H**), contain a germinal vesicle (**H, I**), and possess an abundance of (likely yolk) granules (**I, J**). Blue arrows label germinal vesicles in all relevant panels; yellow arrows label sperm; white arrows label follicular cells. Scale bars: 100 μm (**B**, **C** (inset), **D–F**), 500 μm (**C**), 50 μm (**G–J**).

*Figure 5 continued on next page*

*Figure 5 continued*

The online version of this article includes the following figure supplement(s) for figure 5:

**Figure supplement 1.** Oocytes in *Hofstenia*'s ovaries are surrounded by follicular cells.

*Costello and Costello, 1939*). It has also been suggested that acoels could lay eggs through the mouth; however, direct observation of such egg laying has been challenging (*Apelt, 1969*; *Costello and Costello, 1939*; *Watzin, 1984*).

Previous authors have speculated that *H. miamia* lay eggs through breaks in the body wall, or perhaps partially through the mouth, but egg laying was not observed (*Bock, 1923*; *Steinböck, 1966*). To observe how worms lay eggs, we isolated gravid adult worms and filmed them from underneath for a 24-hr period. Three worms laid a total of 25 eggs in these conditions. We found that worms exclusively laid eggs through their mouths, one at a time, in events typically lasting less than 2 min each (*Figure 6A, B*; *Figure 6—video 1*; *Figure 6—figure supplement 1A*). During this event, the worm performs a series of muscle contractions to transfer a single egg from the ventral medial cavity of fertilized eggs into the pharynx, and then applies further muscle contractions from posterior to anterior to move the egg to the mouth. The worm then places each egg on the substrate with its mouth, likely secreting mucus to attach the egg to the substrate (*Figure 6A, B*; *Figure 6—video 1*). Observations from these videos, as well as of egg clutches in our culturing tanks, show that worms can either deposit a single egg in one location or lay eggs in one or more clutches.

To understand the temporal dynamics of egg laying in *H. miamia*, we then quantified the time course of egg laying in several contexts. We found that adult worms with previous access to mates, once isolated, continued laying eggs for over a month (*Figure 6C*). Juvenile worms reared to adulthood in isolation laid a single burst of eggs (*Figure 6D*), multiple months after first being isolated (although the timing of this burst may depend on rearing conditions and the health of the worms). Consistent with our finding that worms reared entirely in isolation can lay eggs (*Figure 1L, M*), this burst of egg laying is also likely the result of a selfing event in which the worms fertilize their own eggs, or possibly the result of a form of parthenogenesis allowing the activation of unfertilized eggs. We also allowed virgin worms to mate once, in controlled conditions, and subsequently isolated them. These mated and inseminated worms laid eggs for over 3 months after a single mating (*Figure 6E*). One possibility is that these eggs are the result of self-fertilization; however, selfing produces a single burst of egg laying after isolation (*Figure 6D*). Instead, these data considered with our observations of adult ovaries (*Figure 5F, G*) suggest that sperm received during mating may be stored in follicular cells for several months.

Next, we asked how much control the worms had over their egg laying. To test whether worms choose specific locations to lay eggs, we quantified the spatial positions of eggs laid in their culturing tanks. We found that worms have a strong preference for laying eggs on the walls of their tanks, rather than on the floor (*Figure 6F*). Eggs laid on walls are preferentially laid close to the water line (*Figure 6G*, *Figure 6—figure supplement 1B*). This preference is not simply because the worms lay eggs where they are: we observed that worms spend the majority of their time on the floors of their tanks and seem to glide up the walls specifically to lay eggs (*Figure 6—figure supplement 1C*). These spatial preferences do not seem affected by food availability: when deprived of food, egg laying still primarily occurs on the walls of culture tanks (*Figure 6—figure supplement 1D*). These data suggest that worms make active substrate choices for egg laying.

We also observed that worms lay 49% of their eggs in clutches. Individual worms can produce these clutches (*Figure 6—video 1*). Clutch sizes are often small (*Figure 6—figure supplement 1E*), but the largest can contain over 30 eggs deposited within a 3- to 4-day window in culture tanks containing 20–50 worms (*Figure 6H*, *Figure 6—figure supplement 1E*). The vast majority of individual worms lay fewer than 10 eggs in a 3- to 4-day period (*Figure 6—figure supplement 1F*), suggesting that some clutches may be communal. Indeed, in other culturing conditions with large groups of worms, clutch size often exceeds 140 eggs (*Figure 6I*). To test whether worms lay eggs in communal clutches, we allowed worms to lay eggs in tanks for 3 days. We then swapped worms between tanks (or, for control tanks, we swapped worms and removed old eggs). We asked how worms interacted with egg clutches laid in the first 3–4 days and found that many new eggs were laid in pre-existing clutches (*Figure 6J*, *Figure 6—figure supplement 1G*). Worms added new eggs to 42% of old egg clutches. This shows that worms frequently lay eggs in communal egg clutches.

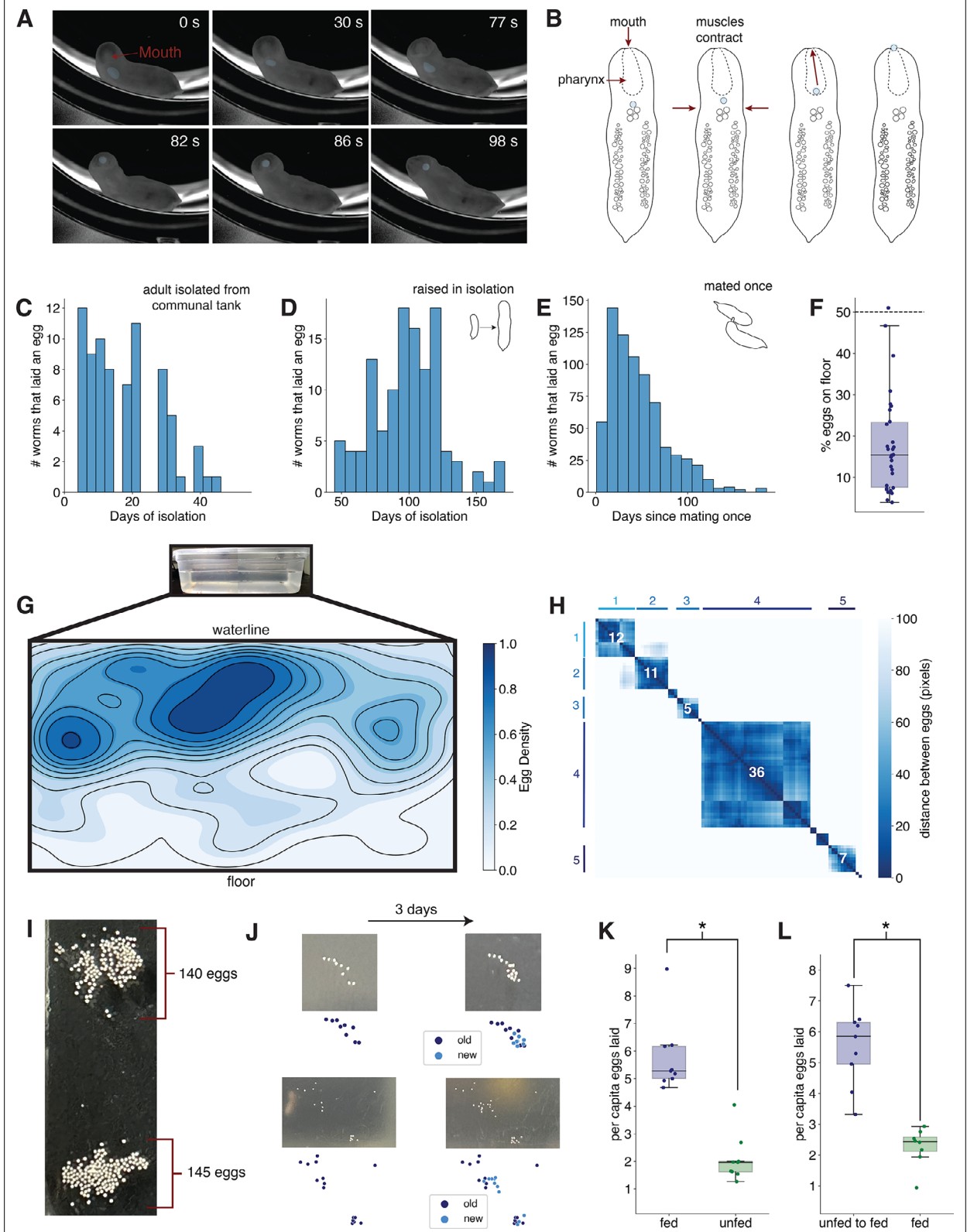

**Figure 6.** *H. miamia* lays eggs through its mouth and exhibits environmental preferences in egg laying. (**A**) Sequence of images from a video of egg laying through the mouth. Eggs in the pharynx and emerging through the mouth are shaded blue. (**B**) Schematic showing presumed process of embryo traveling from the cavity beneath the pharynx to the pharynx and then out through the mouth. (**C**) Histogram showing the timing of eggs laid by adult worms living in communal tanks and then isolated. (**D**) Histogram showing the timing of eggs laid by worms that undergo reproductive development in

*Figure 6 continued on next page*

*Figure 6 continued*

isolation and then self-fertilize. (**E**) Histogram showing the timing of eggs laid by worms that are allowed to mate once. (**F**) Scatterplot of the percentage of eggs found on the floor of communal tanks ($n = 30$). This is significantly different from the expected percentage of eggs based on tank surface area (*t*-test, p < 0.0001). (**G**) Kernel density estimate of egg locations on a subset of tank surfaces with similar dimensions ($n = 2144$ eggs). (**H**) Distance matrix of egg coordinates, with density-based spatial clustering, from a representative tank surface shows that eggs are often laid in clutches. Number of eggs in a clutch is shown in white; cluster identity is shown on the *x* and *y* axes. (**I**) In some culturing conditions, worms lay clutches of up to 145 eggs. (**J**) New worms add eggs to pre-existing clutches laid by other worms. (**K**) Worms that are unfed for 4 days lay fewer eggs than fed worms ($n = 9$ tanks, *t*-test, p < 0.0001). (**L**) Unfed worms that are subsequently fed lay more eggs than worms that are continuously fed ($n \geq 8$ tanks, *t*-test, p < 0.0001).

The online version of this article includes the following video and figure supplement(s) for figure 6:

**Figure supplement 1.** Worms have spatial preferences during egg laying.

**Figure 6—video 1.** Eggs are laid through the mouth.

https://elifesciences.org/articles/105712/figures#fig6video1

Finally, we asked whether worms withhold eggs in suboptimal environments. Our observations (see *Figure 2K, L*) suggested that worms are physiologically resilient to food stress. We therefore conducted an experiment with a set of worms, half of which were deprived of food for one feeding period (3–4 days). The other half were fed normally during this period. We found that food-deprived worms laid significantly fewer eggs during this period (*Figure 6K*). We hypothesized that this reduction in egg laying was because the worms were actively withholding their eggs in the absence of food. Next, we repeated this experiment, first depriving some worms of food and subsequently feeding them. After food-deprived worms were given food, they laid an excess of eggs (*Figure 6L*), showing that the worms withhold egg laying in food-limited environments. More generally, these data show that *H. miamia* assess their environments to decide when and where to lay eggs.

## Discussion

Our work describes the reproductive life history of an acoel across its life cycle (*Figure 7A*). We reveal many new facets of acoel biology, describe the structure and dynamics of growth, regeneration, and egg-laying behavior, and establish a foundation for the experimental study of reproductive biology in acoels. Below, we discuss the significance of our findings for reproductive development, behavior, and the evolution of life history strategies.

### Reproductive development and regeneration

Little is known about the development or regeneration of reproductive structures in acoels. Our data show that in *H. miamia*, reproductive structures develop and regenerate in a stereotyped sequence, with male organs appearing before female ones. During starvation-induced de-growth, these organs degenerate in the opposite sequence. This stereotypy is unexpected and may not be a universal feature of acoels: sparse data from three other acoel species, *Solenofilomorpha funilis* (*Crezée, 1975*), *Otocelis luteola* (*Kozloff, 2000a*), and *Aphanostoma pulchra* (*Perea-Atienza et al., 2013*), indicate that these species do not exhibit identical sequences. The stereotypy we observe is consistent with the idea that a single, size-associated program regulates reproductive organ development in *H. miamia*, and that it may be deployed in a variety of growth contexts.

Our results also show that *H. miamia* is capable of regenerating all reproductive structures from a variety of initial tissue configurations. The trajectories toward organ replacement vary based on this initial state and reveal important features of the regenerative process. For instance, we found that worms missing their sagittal halves or their tails first shrink in size before re-growing. This may be in part because the early stages of wound closure and regeneration require the reorganization of existing tissues. In any case, this reduction in body size is associated with disproportionate reductions in reproductive organ size, demonstrating active tissue destruction mechanisms, possibly mediated by apoptotic processes similar to those reported in other regenerative species (*Ballarin et al., 2008*; *Pellettieri et al., 2010*; *Rychel and Swalla, 2008*). In addition, we found that the two ovaries within a worm grow in a significantly correlated manner, and after sagittal amputation, the missing ovary grows disproportionately to regenerate symmetry. In principle, such bilateral symmetry could be achieved passively, without any feedback (*Reddien, 2018*). However, the strong within-worm correlation, even

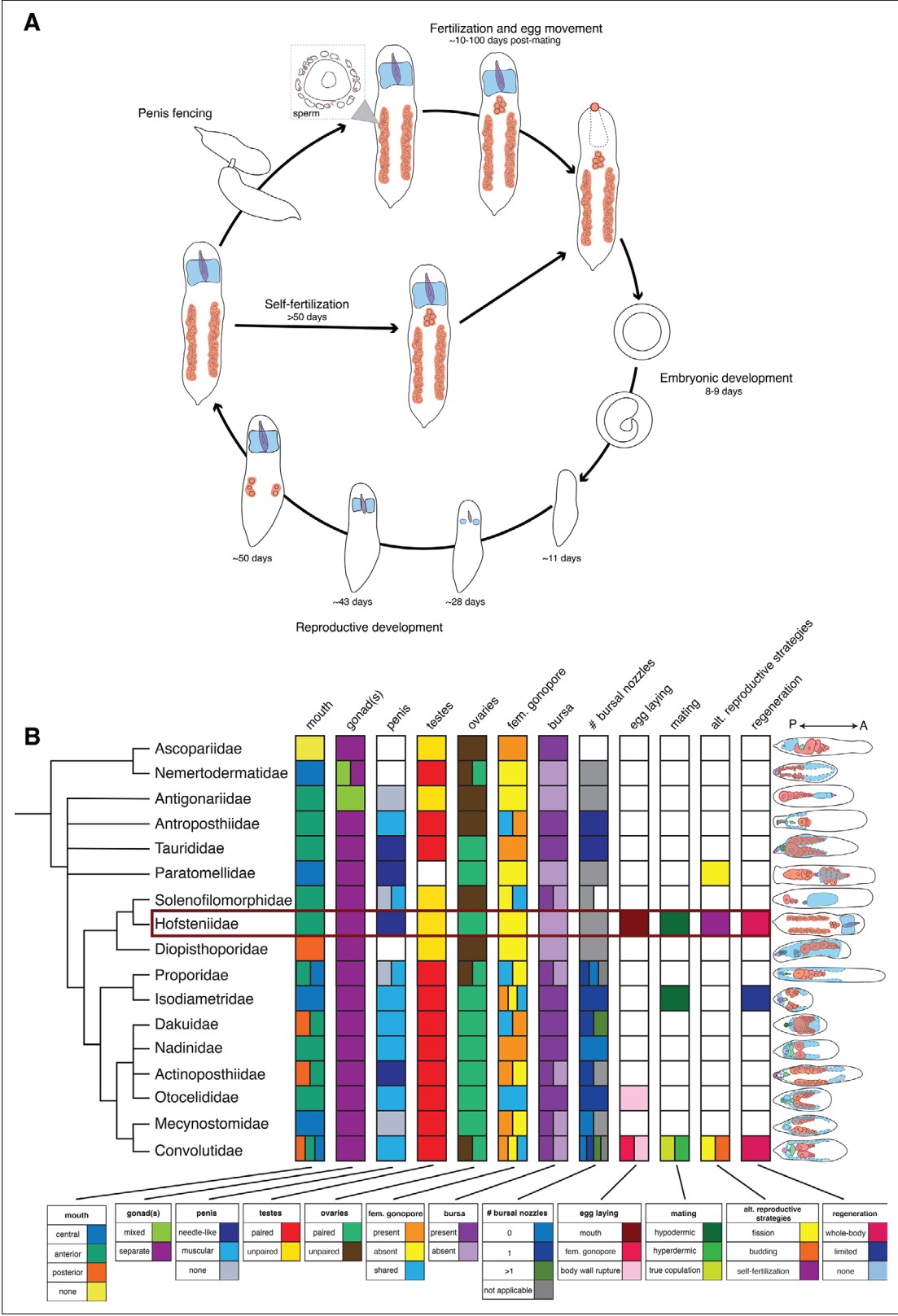

**Figure 7.** Reproductive life histories in Acoelomorpha. (**A**) The life cycle of *Hofstenia miamia*, with major reproductive events displayed. (**B**) Family-level phylogeny of Acoelomorpha (Nemertodermatida, Acoela) showing anatomical and reproductive life history traits (*Supplementary file 1*): position of the mouth, whether gonads are mixed or separated by sex, penis type, paired or unpaired testes, paired or unpaired ovaries, presence or absence of a female gonopore, presence or absence of a seminal bursa, the number of associated bursal nozzles, egg-laying mode, mode of sexual

*Figure 7 continued on next page*

*Figure 7 continued*

reproduction, alternative reproductive strategies, and regenerative capacity (see *Table 3* for definitions of terms and categories). Schematic diagram of the reproductive anatomy of representative species from each family within Acoelomorpha with specific structures colored: male copulatory organ (purple), sperm in testes and/or seminal vesicle (blue), oocytes (red), female or shared gonopore and/or bursa (green) (*Supplementary file 1*). White boxes represent unknown phenotypic states, and in the case of asexual reproduction, its possible absence. Phenotypic classifications are from *Achatz and Hooge, 2006*; *Apelt, 1969*; *Bailly et al., 2014*; *Beltagi and Mandura, 1991*; *Bock, 1923*; *Boone et al., 2011*; *Bush, 1975*; *Costello and Costello, 1939*; *Costello and Costello, 1938*; *Crezee, 1978*; *Dörjes, 1968*; *Dörjes, 1966*; *Faubel, 1976*; *Faubel, 1974*; *Faubel and Cameron, 2001*; *Gardiner, 1895*; *Grae and Kozloff, 1999*; *Hooge, 2003*; *Hooge et al., 2007*; *Hooge and Smith, 2004*; *Hyman, 1937*; *Kostenko, 1989*; *Kozloff, 2000b*; *Meyer-Wachsmuth et al., 2014*; *Peebles, 1915*; *Perea-Atienza et al., 2013*; *Raikova et al., 1995*; *Riser, 1987*; *Shannon and Achatz, 2007*; *Steinböck, 1966*; *Sterrer, 1998*; and *Watzin, 1984*. Phylogeny of Acoelomorpha from *Jondelius et al., 2011* and *Abalde and Jondelius, 2025*.

after controlling for body size, suggests that there is either organism-specific coordination of growth or death rates, or more likely a feedback process that ensures bilateral symmetry.

In many animal lineages such as arthropods, vertebrates, planarians, and nematodes, organs typically grow symmetrically and scale with body size (*Boulan and Léopold, 2021*; *Ko et al., 2024*; *Uppaluri and Brangwynne, 2015*; *Wolpert, 2010*). The mechanisms underlying this coordinated, scaled growth are not fully understood and may vary across tissues and species. For instance, some vertebrate tissues appear to grow using cell-intrinsic programs; others – and many insect tissues – appear to rely on central coordinating mechanisms that read signals of growth from the circulation or from unknown sources (*Allard and Tabin, 2009*; *Boulan and Léopold, 2021*; *Harris et al., 2021*; *Trible and Kronauer, 2017*; *Vallejo et al., 2015*; *Wolpert, 2010*). How different organs ensure appropriate, coordinated scaling remains poorly understood. We find that *H. miamia* must also solve this problem: their reproductive organs scale with body size during development, regeneration, and de-growth, and an active coordination mechanism ensures that the ovaries return to symmetry after perturbation. How do tissues across the body 'read' the same indicators of size, and how is their growth coordinated? Our work establishes a foundation for the experimental study of these and other developmental phenomena.

## Egg-laying physiology and behavior

Few acoels have been directly observed laying eggs (*Costello and Costello, 1939*; *Gardiner, 1895*; *Watzin, 1984*). We find that *H. miamia* lays fertilized eggs through its mouth: a mode of egg laying not observed in other animals. This unusual behavior raises multiple questions. First, where in the animal does fertilization occur? Our observations suggest that follicular cells surrounding immature oocytes may act as a selective barrier to sperm, preventing many sperm cells from reaching the oocyte, a function suggested by *Bock, 1923*. Additionally, given that *H. miamia* can lay eggs for months after a single mating despite lacking a seminal bursa, it is likely that the follicular cells function as a sperm storage organ. Second, we find that *H. miamia* appears capable of self-fertilization as it can lay eggs without mating. While some acoels are capable of asexual reproduction through fissioning and budding (*Åkesson et al., 2001*; *Ax and Schulz, 1959*; *Marcus, 1955*; *Hanson, 1960*; *Ishikawa and Yamasu, 1992*; *Zabotin and Evtugyn, 2021*), self-fertilization has not been reported previously. How does sperm travel to the ovaries? It is likely that sperm cells migrate posteriorly from the testes toward the ovaries in isolated worms, and from arbitrary body regions toward the ovaries after mating. In addition, sperm must travel from all regions of the testes to the seminal and prostatic vesicles. Whether they use chemical cues to navigate toward oocytes and these vesicles, and the mechanics of this process, remain unknown. Third, how do mature eggs travel from the lateral ovaries to the medial cavity, and how are the eggs in this cavity physically loaded into the pharynx? Our anatomical observations have so far failed to reveal obvious oviducts, or other tube-like structures that could facilitate movement of eggs into the pharynx. The only known opening at the posterior of the pharynx is the pharyngeal sphincter that allows food to pass into the gut. We therefore speculate that eggs travel through the gut to the medial cavity, which may simply be a pocket within the gut. Eggs may then be moved into the pharynx through a form of reverse peristalsis (and observation of the worm prior to egg laying shows waves of muscle contraction proceeding posterior-to-anterior as each egg is loaded into the pharynx).

We also find that *H. miamia* appears to make active choices about when and where to lay fertilized eggs. *H. miamia* has clear spatial preferences for egg laying in our culture chambers. Worms often

**Table 3.** Glossary of anatomical and reproductive traits.

**Mouth:** approximate position of the mouth on the ventral surface

| | |
|---|---|
| Central | Mouth located between the anterior end and midpoint of the ventral surface |
| Anterior | Mouth located between the anterior end and midpoint of the ventral surface |
| Posterior | Mouth located between the midpoint of the ventral surface and posterior end |
| None | No mouth |

**Gonad(s):** tissue or organ where gametes are specified

| | |
|---|---|
| Mixed | Sperm and eggs are produced within the same tissue or organ |
| Separate | Sperm and eggs are produced in distinct tissues or organs |

**Penis:** male copulatory organ

| | |
|---|---|
| Needle-like | Penis contains hardened, sclerotized elements |
| Muscular | Absence of hardened elements; soft or glandular penis |
| None | Penis is absent |

**Testes:** tissue where sperm are produced

| | |
|---|---|
| Paired | There are two distinct regions where sperm are produced |
| Unpaired | There is a single region where sperm are produced |

**Ovaries:** tissue where oocytes are produced

| | |
|---|---|
| Paired | There are two distinct regions where oocytes are produced |
| Unpaired | There is a single region where oocytes are produced |

**Fem. gonopore:** female genital opening

| | |
|---|---|
| Present | A female genital opening is present |
| Absent | A female genital opening is not present |

**Bursa:** an organ used to hold sperm received during mating

| | |
|---|---|
| Present | A bursa is present |
| Absent | A bursa is not present |

**# Bursal nozzles:** the number of channels accompanying the bursa; channels that facilitate sperm movement or modification

| | |
|---|---|
| 0 | The bursa is not accompanied by any sperm channels |
| 1 | The bursa has one channel used to move sperm |
| >1 | The bursa has more than one channel used to move sperm |

**Egg laying:** the mode of oviposition that a species has been observed to use

| | |
|---|---|
| Mouth | Eggs are deposited via the mouth |
| fem. gonopore | Eggs are deposited via the female genital opening |
| Body wall rupture | Eggs are deposited via breaks in the body wall |

**Mating:** the mode of sexual reproduction that a species has been observed to use

| | |
|---|---|
| Hypodermic | A copulation mode where sperm is injected beneath the epidermis of a partner |
| Hyperdermal | A copulation mode where sperm are deposited on the epidermis of a partner |
| True copulation | Penis inserted in the female gonopore of a partner; insertion can be mutual or one-sided |

**Alt. reproductive strategies:** forms of reproduction other than two individuals mating

| | |
|---|---|
| Fission | Splitting of an individual into two or more individuals |
| Budding | Development of an outgrowth and then detachment of that outgrowth to form offspring |
| Self-fertilization | Zygote forms from fusion of sperm and egg from a single individual |

lay eggs in communal clutches, sometimes adding to another worm's clutch. They also have environmental preferences, withholding egg laying when deprived of food and then laying these eggs once food becomes available. Together, these results show that *H. miamia* surveys its environment and then integrates this assessment into whether – and where – it is suitable to lay an egg. More generally, the existence of communal egg clutches raises the possibility that acoels may have a rich and unexplored social repertoire.

## Evolution of reproductive strategies within acoels

Placed in a comparative context, our description of *H. miamia*'s reproductive life history reveals that acoels and their sister lineage, the nemertodermatids, display a striking diversity of reproductive morphologies, and likely a corresponding diversity in reproductive behavior (*Figure 7B*). This diversity reveals correlated suites of reproductive traits that suggest a small number of life history strategies. For instance, the presence of a bursa and female gonopore is associated with a muscular penis, suggestive of cooperative, genital-handshake style mating. The absence of a bursa and female gonopore is associated with a needle-laden penis, suggestive of competitive, hypodermic-insemination style mating. Comparative work in the distantly related platyhelminth flatworms shows that these anatomical and behavioral traits indeed coevolve similarly (*Brand et al., 2022a*; *Brand et al., 2022b*), in accordance with social evolutionary theory on sexual conflict (*Charnov, 1979*; *Michiels and Newman, 1998*; *Ramm et al., 2015*). Within the genus *Macrostomum*, a competitive hypodermic insemination syndrome has evolved at least 14 times from ancestral cooperative, reciprocal insemination (*Brand et al., 2022a*; *Brand et al., 2022b*). This syndrome involves the correlated evolution of a sharpened penis, a simplified female sperm-receiving organ, sperm cells lacking bristles and other adaptations for post-copulatory sexual conflict, and associated behavioral changes (*Brand et al., 2022a*; *Brand et al., 2022b*; *Brand et al., 2022c*; *Schärer et al., 2014*; *Schärer et al., 2011*). Whether similar anatomical associations truly predict mating strategies in acoels remains unknown. In addition, the variability of some morphological features does not obviously fit this pattern of cooperative vs. competitive reproductive strategies. For example, the relative locations of the testes and ovaries are highly variable, as is the number of bursal nozzles, suggesting that there may be further evolutionary patterns awaiting explanation. Our work establishes an approach to study reproductive anatomy and behavior in a model acoel. This approach can be applied to many other acoel species. Given their rich diversity, acoels are a promising clade in which to study the evolution of reproductive strategies, and in which to test the generality of theories of sexual conflict.

## Conclusion

These findings establish foundational knowledge of anatomical, physiological, and behavioral elements of *H. miamia*'s reproductive life history. This enables future work on the molecular genetics of reproductive organ development and regeneration, on the physiological processes involved in egg maturation and oviposition, and on the neuroscience of reproductive behavior.

## Methods
### Animal husbandry

We reared gravid *H. miamia* in communal plastic 2.25 l tanks with approximately 1.25 l of artificial seawater (37 ppt, pH 7.8–8.2). These communal tanks were kept in incubators held at 21°C. Twice weekly, we collected embryos, changed their seawater, and fed them with *Artemia* sp. (brine shrimp). The embryos were raised in petri dishes. Once they hatched, we transferred juvenile worms to tanks maintained at room temperature and fed them with marine rotifers (*Brachionus plicatilis*). When reproductive organs began to develop, we transferred worms to larger tanks, each typically housing 30–50 worms. To pair animals for mating, we transferred juvenile worms from tanks to 24-well plates and reared them in isolation following similar feeding and cleaning routines as listed above. In order to study the regeneration of reproductive systems, we isolated adult worms from communal tanks, anesthetized them in 15% tricaine (ethyl 3-aminobenzoate methanesulfonic acid), and then amputated them with micro knives (Fine Science Tools #10316-14). We maintained regenerating worm fragments in 6-well plates and cleaned plates twice weekly. Five days after amputation, we fed these worms *Artemia* sp. (brine shrimp) or rotifers. We also maintained starving animals in 6-well plates that we

cleaned twice weekly but that were not fed. All animals are derived from an inbreeding population of worms collected in 2010 from Bermuda.

## Histology

We fixed adult specimens of *H. miamia* in 4% paraformaldehyde in artificial seawater for 24 hr at room temperature. Following fixation, specimens were washed twice with 70% ethanol to remove the fixative and then preserved in a third change of 70% ethanol for histological preparation. Animals were dehydrated through a graded ethanol series (95% and 100%), cleared in Histoclear, and embedded in Leica Surgipath (Paraplast) with a melting point of 56°C. We made longitudinal, transverse, and sagittal sections at a thickness of 5 μm using a Leitz 1512 microtome. Histological sections were deparaffinized using Histoclear and rehydrated through a graded ethanol series (100%, 95%, 70%, and 50%) to distilled water. Following rehydration, sections were stained using a standard hematoxylin and eosin protocol (*Armed Forces Institute of Pathology, 1960*). Specifically, slides were incubated in Harris Hematoxylin, rinsed with tap water, differentiated in acidic ethanol (70% ethanol with hydrochloric acid), blued in alkaline water (distilled water with ammonium hydroxide), and counterstained with Eosin. After staining, sections were dehydrated through ascending ethanol concentrations, cleared in Histoclear, air-dried overnight, and mounted on glass slides with Permount. Images were acquired using a Zeiss Axio Scan.Z1 and an Olympus BX50 microscope.

## Fluorescence in situ hybridization

We used previously synthesized riboprobes for two genes: *pa1b3-2* and *cgnl1-2* (*Hulett et al., 2023*). Following 1–2 weeks of starvation, we fixed whole worms in 4% paraformaldehyde in artificial sea water for 1 hr at room temperature. We then washed the fixed animals with PBST (PBS + 0.1% Triton X-100) and transferred them to 24-well plates in small baskets with a mesh bottom, with four to six animals in each basket. To remove pigment autofluorescence, we treated the animals with a bleach solution (containing 4% hydrogen peroxide, SSC, and formamide) and left them under a light for 2 hr. For all washes, 800 μl of solution was used. We first permeabilized the animals using proteinase K solution (0.1% SDS, 1 μl/10 ml proK in PBSTx). After 10 min, we post-fixed the worms in 4% formaldehyde in PBST, washed twice in PBST, and then washed in a 1:1 PBST:PreHyb solution for 10 min. We then incubated the samples in PreHyb solution (50% DI formamide, 25% 20X SSC, 0.05–10% Tween-20, 1 mg/ml yeast tRNA, 20% water) for 2 hr in a 56°C hybridization oven and then transferred them to a hybridization solution (50% DI formamide, 25% 20X SSC, 0.05–10% Tween-20, 1 mg/ml yeast tRNA, 10–50% dextran sulfate, 10% water) containing riboprobe(s) overnight. We then put the specimens through several 30 min washes at 56°C, starting with two PreHyb washes, two 1:1 PreHyb:2XSSCT washes, two 2XSSCT (2XSSC with 0.1% Triton X-100) solution washes, two 0.2XSSCT (0.2X SSC with 0.1% Triton X-100) solution washes. We cooled the specimens to room temperature and then washed them in two 10-min PBST washes. We then performed a 1-hr blocking at room temperature with blocking solution (5% horse serum and 5% casein in PBST). We incubated specimens with anti-Digoxigenin-POD (1:1500 dilution; Roche, 11633716001) in a blocking buffer overnight. The following day, we washed animals six times with PBST, incubated with a tyramide buffer (1.1688 g/ml NaCl, 6.18 mg/ml boric acid, filled with water and adjusted pH to 8.5) for 10 min, then developed with rhodamine-conjugated tyramide solution for 10 min. We washed animals again with PBST and incubated for 1 hr in 1:500 PBST:Hoechst to label nuclei. We mounted the animals on glass slides with VECTASHIELD PLUS Antifade Mounting Medium (Vector Laboratories, H-1900). We estimated worm length (*Figure 4A–D*) by measuring copulatory apparatus length (gonopore to prostatic vesicle) or cross-sectional worm width within the image and comparing that measurement to measurements of developing worms.

## Whole-mount immunofluorescence

We fixed worms in 4% paraformaldehyde (Electron Microscopy Sciences, 15714) in artificial sea water for 1–2 hr at room temperature (juveniles were fixed for 1 hr, adults were fixed for two) before washing with PBST (PBS + 0.1% Triton X-100). We anesthetized adult worms in 0.5 mg/ml tricaine for 5 min prior to fixation to minimize epidermal rupture. For immunofluorescence, we washed worms in PBST, blocked for 1 hr at room temperature in 10% goat serum in PBST, and incubated them in primary antibody for 48–72 hr at 4°C on a shaker (juveniles for 48 hr, adults for 72 hr). The following

day, we washed worms thoroughly in PBST (8 × 20 min washes) on a nutator before blocking for 1 hr at room temperature on a shaker. We incubated worms in secondary antibody overnight at 4°C; the following day, we washed them thoroughly in PBST (8 × 20 in washes) on a shaker before adding direct conjugate dyes (Hoechst (Thermo Fisher, H1399), SiR-actin (Cytoskeleton, CY-SC001)) and mounting them on glass slides with VECTASHIELD HardSet Antifade Mounting Medium (Vector Laboratories, H-1400-10). We used the following antibodies: Tropomyosin (custom) (*Hulett et al., 2020*), Piwi-1 (custom), Goat anti-Rabbit IgG (H+L) Cross-Adsorbed Secondary Antibody, Alexa Fluor 568 (Thermo Fisher, A-11011), and FMRFamide (EMDMillipore AB15348). These antibodies were selected because, in previous work, we found that they labeled relevant morphological structures in *Hofstenia*. We acquired all images with FISH and immunofluorescence on a Leica SP8 point-scanning confocal microscope. We generated the Piwi-1 (Rabbit polyclonal) custom antibody using GenScript as previously described for Tropomyosin (*Hulett et al., 2020*). Briefly, a peptide was synthesized from the *H. miamia* Piwi-1 protein sequence, expressed in *E. coli*, and used to immunize rabbits. The Piwi-1 antibody used in this paper was #SC1195 (0.845 mg/ml), used at a concentration of 2 µg/ml.

## Quantitative analysis of development, regeneration, and de-growth

We collected 42 zygotes laid from wild-type worms. Two embryos did not hatch, and two worms died during the course of the experiment (likely due to handling error). Four worms were removed from the experiment at different time points for additional imaging, resulting in 34 worms that we followed into adulthood (*Figure 1—figure supplement 2B*). We transferred each embryo to a well of a 24-well cell culture plate with artificial seawater. Plates were stored in a temperature-controlled incubator. Without removing the embryos from the plate, we imaged them twice weekly through a dissection microscope with white illumination from LEDs mounted above the sample. We also changed water on these days. We added the same volume of water to each well (2.35 ml in each 12-well). Once at least one embryo had hatched, we added rotifers to each well. On each feeding day, we calculated the concentration of rotifers by counting the number of rotifers in diluted samples. Based on this concentration, we changed the volume of rotifers added to each well to ensure a consistent number of rotifers was given to each animal. At different stages during the experiment, we increased the number of rotifers (*Figure 1—figure supplement 2A*). These increases were gradual to ensure worms always had more food than they could eat, while minimizing excess food which could affect water quality. We removed hatched worms from wells for imaging twice weekly. We anesthetized worms in tricaine and mounted them on microscope slides with a 20 × 20 mm sticker grid with 1 mm resolution (Thomas Scientific, #1207X53). Depending on the worms' pigmentation, this grid is visible through the body of some worms, but this did not impact the visibility of reproductive organs. When worms were larger, we moved them to larger 6-well plates in 10 ml of artificial sea water and began to feed them brine shrimp in addition to rotifers (*Figure 1—figure supplement 2A*). We measured the number of brine shrimp added using the same concentration procedure outlined above for rotifers. Once worms had matured, we increased the number of brine shrimp.

We manually annotated images in FIJI. We measured lengths from the most posterior part of the male reproductive system (e.g., base of the penis tube before sperm production begins or base of the seminal vesicle) to the male gonopore, the length of the seminal vesicle, the length of each ovary, the length of the worm's body, and the width of the worm's body. We calculated penis length by subtracting the length of the seminal vesicle from the length of the male gonad. All measurements were converted to millimeters using image-specific scale information.

We repeated these imaging and measurement processing procedures with starving worms and regenerating worm fragments. We anesthetized and imaged starved and fed control worms roughly once a week for 11 weeks. 106 days after isolation, we imaged the starved worms again, at which point one starved worm had died. We imaged regenerating worms starting 2 days post-amputation and then again every 2–5 days until 25 days post-amputation, at which point we took images less frequently. We did not feed regenerating fragments for the first 4 days after amputation and then fed them normally. In these experiments, we used a two-pointed line or multi-jointed line to measure lengths. We performed all analyses in Python, with code assistance from GPT4.

## Egg-laying behavior

We moved adult worms from a communal tank to individual wells in the center of a 24- or 12-well plate (Fisher Scientific #07-200-82). We mounted the plate on an elevated platform within an enclosed behavioral rig. The plate was illuminated with uniform white light, from addressable LEDs (Adafruit #SK6812RGBW, powered by 5V DC) arranged in a ring around the plate behind a white acrylic diffuser (McMaster-Carr #8505K741). An infrared camera (Basler Ace ac4024-29um USB 3.0 monochrome) was mounted underneath the plate to allow filming of the worm's ventral surface. A USB fan was mounted on the floor of the behavioral chamber for temperature regulation. We filmed worms at 1 Hz for 8–12 hr. After filming, each worm was returned to its original communal tank.

To determine the timing of egg laying in adults with continuous access to mates, we collected adult worms from communal tanks, and each was placed in an individual well of a 6-well plate. Every 3–4 days, we counted the number of eggs laid by each worm, removed the eggs, changed the water, and fed worms with brine shrimp or rotifers. To induce self-fertilization, we moved juvenile worms with immature reproductive systems from communal tanks to 24-well plates and fed them with rotifers. Once these worms had increased in size, we moved them to 6-well plates and fed them with artemia. We recorded egg counts every 3–4 days when water and food were refreshed. To determine the time course of egg laying after mating once, we isolated juvenile worms. Once these worms were reproductively mature, we paired each worm with another isolated worm and allowed them to mate. After mating, we returned the worms to isolation. We recorded egg counts as previously described. If worms stopped laying eggs for more than 2 weeks, we allowed them to mate again before re-isolating them.

To find egg positions, we maintained communal tanks according to our husbandry protocol. Every 3–4 days, before cleaning, we recorded images of each face of the tank (two short faces, two long faces, and the floor of the tank). We then scored the x- and y-coordinates of eggs on each face using FIJI. Since few eggs were laid on the floors of the tanks and it was unclear whether these eggs were attached or had been dislodged from the walls of the tanks, we did not score their locations (*Figure 6F*). We also scored the x- and y-coordinates of the boundaries of the water in each image. From these image-specific boundaries, we calculated a common set of bounds for each type of face (i.e., short and long). We scaled egg coordinate positions and aligned them to common bounds for analysis of spatial egg laying preferences. To ask whether worms add to existing clutches of eggs, we cared for communal tanks as previously described but did not remove eggs from their faces. We took images of each face, and then we transferred each cohort of worms in a tank to a different tank. After 3–4 days, we took images of tank faces again, scored coordinates of each egg, and recorded whether each egg was old or newly laid.

For experiments measuring how worms altered their egg-laying behavior based on food availability, we selected 17 and 18 communal worm tanks, respectively. On the first day of the experiment, half of these tanks were randomly selected to have food (brine shrimp) withheld. After 4 days with or without food, we collected and counted eggs from all tanks. To test whether egg laying recovers when unfed worms are fed, we randomly selected communal tanks of adult worms to have food withheld. Four days after food was added or withheld, we removed all eggs that were laid and then fed all communal tanks. Three days later, eggs were collected and counted.

## Acknowledgements

We thank Gonzalo Giribet, Matthew Hooge, James Hanken, Javier Ortega-Hernandez, Andrew Berry, and all members of the Srivastava lab for helpful discussion. We also thank James Hanken, Júlia Chaumel, and Christina Daly for histology help and infrastructure. VC and APK are Fellows of the Jane Coffin Childs Fund for Medical Research. ST acknowledges funding from the Harvard Museum of Comparative Zoology, Herchel Smith Undergraduate Science Research Program, and the Program for Research in Science and Engineering. This project was supported by grants from the National Institute of General Medical Sciences of the NIH to MS: R35GM128817 and R35GM153252.

## Additional information

### Funding

| Funder | Grant reference number | Author |
|---|---|---|
| National Institute of General Medical Sciences | R35GM128817 | Mansi Srivastava |
| National Institute of General Medical Sciences | R35GM153252 | Mansi Srivastava |
| Harvard Museum of Comparative Zoology | Herchel Smith Undergraduate Science Research Program | Samantha Elizabeth Tseng |
| Jane Coffin Childs Fund for Medical Research | | Vikram Chandra Allison P Kann |

The funders had no role in study design, data collection, and interpretation, or the decision to submit the work for publication.

### Author contributions

Vikram Chandra, Conceptualization, Data curation, Software, Formal analysis, Supervision, Funding acquisition, Validation, Investigation, Visualization, Methodology, Writing – original draft, Writing – review and editing; Samantha Elizabeth Tseng, Software, Formal analysis, Funding acquisition, Validation, Investigation, Visualization, Methodology, Writing – original draft, Writing – review and editing; Allison P Kann, Formal analysis, Validation, Investigation, Visualization, Methodology, Writing – review and editing; Diana Marcela Bolanos, Validation, Investigation, Visualization, Methodology, Writing – review and editing; Mansi Srivastava, Conceptualization, Supervision, Funding acquisition, Writing – original draft, Project administration, Writing – review and editing

### Author ORCIDs

Vikram Chandra ⓘ https://orcid.org/0000-0002-8020-0664
Samantha Elizabeth Tseng ⓘ https://orcid.org/0009-0002-6683-8167
Allison P Kann ⓘ https://orcid.org/0000-0003-0111-9081
Diana Marcela Bolanos ⓘ https://orcid.org/0000-0003-3236-0062
Mansi Srivastava ⓘ https://orcid.org/0000-0002-2126-8634

Reviewer #1 (Public review): https://doi.org/10.7554/eLife.105712.3.sa1
Reviewer #2 (Public review): https://doi.org/10.7554/eLife.105712.3.sa2
Author response https://doi.org/10.7554/eLife.105712.3.sa3

## Additional files

### Supplementary files

Supplementary file 1. Survey of reproductive traits within Acoelomorpha. see attached.xlsx file.

MDAR checklist

### Data availability

All data and code for quantitative analyses in this manuscript are available at Zenodo (https://zenodo.org/records/16923213).

The following dataset was generated:

| Author(s) | Year | Dataset title | Dataset URL | Database and Identifier |
|---|---|---|---|---|
| Tseng S, Chandra V | 2025 | Data and code: Developmental, regenerative, and behavioral dynamics in acoel reproduction | https://doi.org/10.5281/zenodo.16923213 | Zenodo, 10.5281/zenodo.16923213 |

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
