## [Editor Report · eLife Assessment]

Xenacoelomorpha is an enigmatic phylum, displaying various presumably simple or ancestral bilaterian features. This **valuable** study characterises the reproductive life history of Hofstenia miamia, a member of class Acoela in this phylum. The authors describe the morphology and development of the reproductive system, its changes upon degrowth and regeneration, and the animals' egg-laying behaviour. The evidence is **convincing**, with fluorescent microscopy and quantitative measurements as a considerable improvement to historical reports based mostly on histology and qualitative observations.

---

## [Referee Report · Reviewer #1 (Public review)]

The aim of this study was a better understanding of the reproductive life history of acoels. The acoel Hofstenia miamia, an emerging model organism, is investigated; the authors nevertheless acknowledge and address the high variability in reproductive morphology and strategies within Acoela.

The morphology of male and female reproductive organs in these hermaphroditic worms is characterised through stereo microscopy, immunohistochemistry, histology, and fluorescent in situ hybridization. The findings confirm and better detail historical descriptions. A novelty in the field is the in situ hybridization experiments, which link already published single-cell sequencing data to the worms' morphology. An interesting finding, though not further discussed by the authors, is that the known germline markers cgnl1-2 and Piwi-1 are only localized in the ovaries and not in the testes.

The work also clarifies the timing and order of appearance of reproductive organs during development and regeneration, as well as the changes upon de-growth. It shows an association of reproductive organ growth to whole body size, which will be surely taken into account and further explored in future acoel studies. This is also the first instance of non-anecdotal degrowth upon starvation in H. miamia (and to my knowledge in acoels, except recorded weight upon starvation in Convolutriloba retrogemma [1]).

Egg laying through the mouth is described in H. miamia for the first time as well as the worms' behavior in egg laying, i.e. choosing the tanks' walls rather than its floor, laying eggs in clutches, and delaying egg-laying during food deprivation. Self-fertilization is also reported for the first time.

The main strength of this study is that it expands previous knowledge on the reproductive life history traits in H. miamia and it lays the foundation for future studies on how these traits are affected by various factors, as well as for comparative studies within acoels. As highlighted above, many phenomena are addressed in a rigorous and/or quantitative way for the first time. This can be considered the start of a novel approach to reproductive studies in acoels, as the authors suggest in the conclusion. It can be also interpreted as a testimony of how an established model system can benefit the study of an understudied animal group.

The main weakness of the work is the lack of convincing explanations on the dynamics of self-fertilization, sperm storage, and movement of oocytes from the ovaries to the central cavity and subsequently to the pharynx. These questions are also raised by the authors themselves in the discussion. Another weakness (or rather missing potential strength) is the limited focus on genes. Given the presence of the single-cell sequencing atlas and established methods for in situ hybridization and even transgenesis in H. miamia, this model provides a unique opportunity to investigate germline genes in acoels and their role in development, regeneration, and degrowth. It should also be noted that employing Transmission Electron Microscopy would have enabled a more detailed comparison with other acoels, since ultrastructural studies of reproductive organs have been published for other species (cfr e.g. [2],[3],[4]). This is especially true for a better understanding of the relation between sperm axoneme and flagellum (mentioned in the Results section), as well as of sexual conflict (mentioned in the Discussion).

(1) Shannon, Thomas. 2007. 'Photosmoregulation: Evidence of Host Behavioral Photoregulation of an Algal Endosymbiont by the Acoel Convolutriloba Retrogemma as a Means of Non-Metabolic Osmoregulation'. Athens, Georgia: University of Georgia [Dissertation].

(2) Zabotin, Ya. I., and A. I. Golubev. 2014. 'Ultrastructure of Oocytes and Female Copulatory Organs of Acoela'. Biology Bulletin 41 (9): 722-35.

(3) Achatz, Johannes Georg, Matthew Hooge, Andreas Wallberg, Ulf Jondelius, and Seth Tyler. 2010. 'Systematic Revision of Acoels with 9+0 Sperm Ultrastructure (Convolutida) and the Influence of Sexual Conflict on Morphology'.

(4) Petrov, Anatoly, Matthew Hooge, and Seth Tyler. 2006. 'Comparative Morphology of the Bursal Nozzles in Acoels (Acoela, Acoelomorpha)'. Journal of Morphology 267 (5): 634-48.

---

## [Referee Report · Reviewer #2 (Public review)]

Summary:

While the phylogenetic position of Acoels (and Xenacoelomorpha) remains still debated, investigations of various representative species are critical to understanding their overall biology.

Hofstenia is an Acoels species that can be maintained in laboratory conditions and for which several critical techniques are available. The current manuscript provides a comprehensive and widely descriptive investigation of the productive system of Hofstenia miamia.

Strengths:

(1) Xenacoelomorpha is a wide group of animals comprising three major clades and several hundred species, yet they are widely understudied. A comprehensive state-of-the-art analysis on the reprodutive system of Hofstenia as representative is thus highly relevant.

(2) The investigations are overall very thorough, well documented, and nicely visualised in an array of figures. In some way, I particularly enjoyed seeing data displayed in a visually appealing quantitative or semi-quantitative fashion.

(3) The data provided is diverse and rich. For instance, the behavioral investigations open up new avenues for further in-depth projects.

Weaknesses:

While the analyses are extensive, they appear in some way a little uni-dimensional. For instance the two markers used were characterized in a recent scRNAseq data-set of the Srivastava lab. One might have expected slightly deeper molecular analyses. Along the same line, particularly the modes of spermatogenesis or oogenesis have not been further analysed, nor the proposed mode of sperm-storage.

[Editors' note: In their response, the authors have suitably addressed these concerns or have satisfactorily explained the challenges in addressing them.]

---

## [Author Response]

The following is the authors’ response to the original reviews.

**Reviewer #1 (Recommendations for the authors):**
I will address here just some minor changes that would improve understanding, reproducibility, or cohesion with the literature.(1) It would be good to mention that the prostatic vesicle of this study is named vesicula granulorum in (Steniböck, 1966) and granule vesicle in (Hooge et al, 2007).

We have now included this (line 90 of our revised manuscript).

(2) A slightly more detailed discussion of the germline genes would be interesting. For example, a potential function of pa1b3-2 and cgnl1-2 based on the similarity to known genes or on the conserved domains.

Pa1b3-2 appears to encode an acetylhydrolase; cgnl1-2 is likely a cingulin family protein involved in cell junctions. However, given the evolutionary distance between acoels and model organisms in whom these genes have been studied, we believe it is premature to speculate on their function without substantial additional work. We believe this work would be more appropriate in a future publication focused on the molecular genetic underpinnings of Hofstenia’s reproductive systems and their development.

(3) It is mentioned that the animals can store sperm while lacking a seminal bursa "given that H. miamia can lay eggs for months after a single mating" (line 635) - this could also be self-fertilization, according to the authors' other findings.

We agree that it is possible this is self-fertilization, and we believe we have represented this uncertainty accurately in the text. However, we do not think this is likely, because self-fertilization manifests as a single burst of egg laying (Fig. 6D). We discuss this in the Results (line 540).

(4) A source should be given for the tree in Figure 7B.

We have now included this source (line 736), and we apologize for the oversight.

(5) Either in the Methods or in the Results section, it would be good to give more details on why actin and FMRFamide and tropomyosin are chosen for the immunohistochemistry studies.

We have now included more detail in the Methods (line 823). Briefly, these are previously-validated antibodies that we knew would label relevant morphology.

(6) In the Methods "a standard protocol hematoxylin eosin" is mentioned. Even if this is a fairly common technique, more details or a reference should be provided.

We have now included more detail, and a reference (lines 766-774).

(7) Given the historical placement of Acoela within Platyhelminthes and the fact that the readers might not be very familiar with this group of animals, two passages can be confusing: line 499 and lines 674-678.

We have edited these sentences to clarify when we mean platyhelminthes, which addresses this confusion.

(8) A small addition to Table S1: Amphiscolops langerhansi also presents asexual reproduction through fission ([1], cited in [2]).

Thanks. We have included this in Table S1.

(a) Hanson, E. D. 1960. 'Asexual Reproduction in Acoelous Turbellaria'. The Yale Journal of Biology and Medicine 33 (2): 107-11.

(b) Hendelberg, Jan, and Bertil Åkesson. 1991. 'Studies of the Budding Process in Convolutriloba Retrogemma (Acoela, Platyhelminthes)'. In Turbellarian Biology: Proceedings of the Sixth International Symposium on the Biology of the Turbellaria, Held at Hirosaki, Japan, 7-12 August 1990, 11-17. Springer.

**Reviewer #2 (Recommendations for the authors):**
I do not have any major comments on the manuscript. By default, I feel descriptive studies are a critical part of the advancement of science, particularly if the data are of great quality - as is the case here. The manuscript addresses various topics and describes these adequately. My minor point would be that in some sections, it feels like one could have gone a bit deeper. I highlighted three examples in the weakness section above (deeper analysis of markers for germline; modes of oogenesis/spermatogenesis; or proposed model for sperm storage). For instance, ultrastructural data might have been informative. But as said, I don't see this as a major problem, more a "would have been nice to see".

We have responded to these points in detail above.